



# Simulated or measured soil moisture: Which one is adding more value to regional landslide early warning?

Adrian Wicki[1], Per-Erik Jansson[2], Peter Lehmann[3], Christian Hauck[4], Manfred Stähli[1]

[1]Swiss Federal Institute for Forest, Snow and Landscape Research WSL, Zürcherstrasse 111, 8903 Birmensdorf, Switzerland
[2]KTH Royal Institute of Technology, 100 44 Stockholm, Sweden
[3]ETH Zurich, Institute of Terrestrial Ecosystems, Universitätstrasse 16, 8092 Zürich, Switzerland
[4]University of Fribourg, Department of Geosciences, Chemin du Musée 4, 1700 Fribourg, Switzerland

*Correspondence to*: Adrian Wicki (adrian.wicki@wsl.ch)

**Abstract.** The inclusion of soil wetness information in empirical landslide prediction models was shown to improve the
forecast goodness of regional landslide early warning systems (LEWS). However, it is still unclear which source of information – numerical models or in-situ measurements – are of higher value for this purpose. In this study, soil moisture dynamics at 133 grassland sites in Switzerland were simulated for the period of 1981 to 2019 using a physically-based 1D soil moisture transfer model (CoupModel). A common parametrization set was defined for all sites except for site-specific soil hydrological properties, and the model performance was assessed at a subset of 14 sites where in-situ soil moisture measurements were
available on the same plot. A previously developed statistical framework was applied to fit an empirical landslide forecast model, and ROC analysis was used to assess the forecast goodness. To assess the sensitivity of the landslide forecasts, the statistical framework was applied to different CoupModel parametrizations, to various distances between simulation sites and landslides, and to measured soil moisture from a subset of 35 sites for comparison with a measurement-based forecast model. We found that (i) simulated soil moisture is a skilful predictor for regional landslide activity, (ii) that it is sensitive to the
formulation of the upper and lower boundary conditions, and (iii) that the information content is strongly distance-dependent. Compared to a measurement-based landslide forecast model, the model-based forecast performs better as the homogenization of hydrological processes and the site representation can lead to a better representation of triggering event conditions. However, it is limited in reproducing critical antecedent saturation conditions due to an inadequate representation of the long-term water storage.

## 1 Introduction

Landslides are a major natural hazard causing fatalities and damages in mountainous regions worldwide (Froude and Petley, 2018). The term "landslide" includes various types of mass movements spanning over different source materials (e.g. soil, rock), process dynamics (e.g. slide, flow, fall) and trigger types (e.g. water infiltration, earthquakes, human interaction) (Hungr et al., 2014; Varnes, 1978; Wieczorek, 1996). Here, we focus on rainfall and snow-melt triggered shallow landslides which
occur frequently in Switzerland (Hess et al., 2014). The landslide process can be analysed by cause and trigger factors (Bogaard



and Greco, 2016): Factors that precondition the slope to sliding ("cause factors") include the long-term weathering of the slope material, the topographic disposition, the characteristics of the vegetation cover, and the hydrological prewetting of the slope. The eventual failure of a slope along a shear plane is connected to a local and short-duration decrease in shear strength ("trigger") due to pore-water pressure increase from direct rainfall or snow-melt water infiltration or due to the indirect build-up of a perched water table or groundwater table (Bogaard and Greco, 2016; Terlien, 1998; Terzaghi, 1943).

Risk awareness and the corresponding response of people is a significant factor for mortality particularly during shallow landslide events (Pollock and Wartman, 2020). In this respect, landslide early warning systems (LEWS), which allow the prediction of the landslide danger, have become an essential part of risk management in many places around the world (e.g. Baum et al., 2010; Guzzetti et al., 2020; Stähli et al., 2015). Regional LEWS, also referred to as territorial (Piciullo et al., 2018) or geographical (Guzzetti et al., 2020) LEWS, make predictions for multiple landslides and operate at regional to national scale. Statistical landslide forecast models relate environmental variables such as rainfall characteristics or soil wetness variation to the occurrence of landslides. They are fundamentally based on time series of environmental data and a comprehensive landslide inventory (Guzzetti et al., 2020; Terlien, 1998).

In the past, many regional LEWS have been based on statistical forecast models that describe empirical relationships between rainfall events and landslide occurrence (Caine, 1980; Guzzetti et al., 2008; Segoni et al., 2018a). While this approach benefits from widely available rainfall data, the focus on triggering factors disregards the influence of the antecedent wetness conditions ("cause factors"), which could be represented by including soil wetness information (Bogaard and Greco, 2018). In fact, forecast goodness improvement was reported after incorporation of in situ soil moisture measurements (Mirus et al., 2018a, 2018b; Thomas et al., 2020), remotely sensed soil moisture (Bordoni et al., 2020; Brocca et al., 2016; Thomas et al., 2019; Zhao et al., 2019a; Zhuo et al., 2019b) or simulated soil moisture using physically-based models (e.g. Ponziani et al., 2012; Segoni et al., 2018b; Zhuo et al., 2019a). Other landslide forecast models exist that express antecedent wetness conditions in terms of accumulated pre-event precipitation (e.g. Aleotti, 2004; Martelloni et al., 2012), or antecedent soil wetness or precipitation indices (e.g. Crozier, 1999; Glade, 2000; Godt et al., 2006).

At the point scale, in situ soil moisture sensors (time- and frequency domain reflectometry, TDR, FDR, or capacitance based) estimate dielectric permittivity (Babaeian et al., 2019) from which soil moisture is deduced using an empirical calibration function (for example equation of Topp et al., 1980). They are representative for a specific volume of soil and are usually integrated to depth profiles. While sensor networks deliver soil moisture estimates at high temporal resolution, installation and long-term maintenance are costly and difficult, and the representativeness for regional landslide activity decreases significantly with distance from the soil moisture site (Wicki et al., 2020). Larger spatial integration is achieved by using remotely sensed soil moisture information derived from microwave emissions (Reichle et al., 2017). However, the spatial and temporal resolution are coarse and the sensing depth is shallow, limiting the potential for LEWS applications in mountainous regions (Thomas et al., 2019; Zhuo et al., 2019a).

Numerical models for the simulation of soil water dynamics may help in this regard as they provide cheap, continuous and spatially and temporally consistent soil moisture estimates. Such models typically simulate the accumulation and redistribution



of water (and heat) either in specific soil profiles (in one dimension) or for larger areas (pixels or hydrological response units) for time resolutions from minutes to days. Physically-based models explicitly represent hydrologic state variables and fluxes by mathematical formulations (Fatichi et al., 2016), where the variably saturated water flow is often described by the Richards' equation (1931) and the mathematical expressions in form of partial differential equations are solved with a numerical method (Feddes et al., 1988). In comparison to simpler conceptual or bucket models, physically-based models are more time-

consuming in calculation and require more parameter settings. However, they are less dependent on specific calibration procedures, since parameter values can be constrained by observable quantities or expert decisions (Gharari et al., 2014; Gupta and Nearing, 2014), or they can be inferred from easier measurable quantities by means of pedotransfer functions (Van Looy et al., 2017; Schaap et al., 2001). One-dimensional coupled water and heat transfer models go back to the pioneering work of Harlan (1973) and were further developed and implemented in computer codes for example by van Genuchten (WORM, 1987),

Jansson (CoupModel, 2012) or Šimůnek et al. (HYDRUS-1D, 2012). Two-dimensional soil hydrological models, such as PREVAH (Viviroli et al., 2009), WaSiM-ETH (Klok et al., 2001), TOPKAPI (Liu and Todini, 2002) or Tethys-Chloris (Fatichi et al., 2012) to name a few, are typically applied at catchment or regional scale. Due to the larger coverage, they are restricted by computational resources and often have to simplify the modelling process (e.g. by reducing the temporal resolution, the number of modelling layers or the number of processes represented), but they have the advantage of lateral connectivity and

basin-wide coverage (Fatichi et al., 2016). Common limitations of all physically-based models are mainly related to the availability of appropriate soil physical properties to describe the soil hydraulic characteristics, simplifications of the model boundary conditions and the mathematical description of the hydrological processes, and the quality of the dynamic input data (Feddes et al., 1988; Paniconi and Putti, 2015).

Ultimately, the question arises to what extent landslide forecast models that are based on simulated soil moisture are reliable

and representative in comparison to models based on actual soil moisture measurements. In this study, we aim (i) to clarify the skill of a LEWS based on simulated soil moisture compared to one based on soil moisture measurements, (ii) to assess the sensitivity of this skill to model assumptions and parameters, and (iii) to evaluate the potential of extending a measurement-based LEWS to sites with no soil moisture measurements. This study assesses the potential and limitations of using a soil water transfer model for regional landslide early warning and highlights the strengths and weaknesses compared to using soil

moisture measurements.

## 2 Material and methods

### 2.1 Study design

The following section summarizes the design of this study: In a first step, soil moisture was simulated at 133 sites in Switzerland using a 1D soil moisture transfer model. Second, the forecast goodness for regional landslide activity was assessed

by fitting and evaluating a statistical landslide forecast model to observed shallow landslides. Finally, the landslide forecast goodness was compared with a landslide forecast model based on in situ soil moisture measurements available at a subset of





the modelled sites. We used a statistical framework previously developed to assess the information content of in situ soil wetness information for regional landslide activity (Wicki et al., 2020), and we used the same soil moisture monitoring dataset compiled in the named study for comparison with a measurement-based forecast model. We focused on in situ information
and 1D modelling because of the high temporal resolution and in-depth integration, and due to the availability of 1D validation data.

**2.2 Soil moisture model**

In this study, the heat and mass transfer model CoupModel (P.-E. Jansson, 2012) was used to simulate soil water transfer along a 1D virtual soil profile. The CoupModel has been used extensively to simulate temporal soil moisture dynamics (e.g. Okkonen
et al., 2017; Pellet et al., 2016; Scherler et al., 2010; Wu et al., 2020; Wu and Jansson, 2013) and soil water balance variations (e.g. Christiansen et al., 2006; Madani et al., 2018; Walthert et al., 2015). In the context of landslide early warning, parts of the CoupModel were used for soil moisture simulations within the Norwegian national forecasting service for predicting rainfall-induced landslides (Krøgli et al., 2018).

At the core of the model, two coupled differential equations for water and heat transport are solved assuming that flows are
the result of gradients (Jansson and Karlberg, 2011). The soil water flow, $q_w$, follows Darcy's law as generalised for unsaturated flow by Buckingham (1907)

$$q_w = -K_w \left( \frac{\delta \psi}{\delta z} - 1 \right), \tag{1}$$

where $K_w$ is the unsaturated hydraulic conductivity, $\psi$ is the matric potential head, and z is the depth. Formulations to simulate vapour flow and bypass flow in macropores are implemented in the CoupModel as well but were not included in this study.
From Eq. (1) and the law of mass conservation, the unsaturated water flow equation follows

$$\frac{\partial \theta}{\partial t} = -\frac{\partial q_w}{\partial z} + s_w, \tag{2}$$

where $\theta$ is the soil water content and $s_w$ is a source or sink term.

To solve the water flow equation, two soil characteristic hydraulic properties need to be defined for each model layer, both of which are considered to be functions of the water content: the soil water retention curve characterizing the relationship between
matric potential and water content and the unsaturated hydraulic conductivity function, describing the saturated conductivity as function of water saturation (or matric potential) and the saturated soil hydraulic conductivity. In this study, they were defined by the Mualem and van Genuchten closed-form equations (van Genuchten, 1980; Mualem, 1976)

$$\theta = \theta_r + \frac{\theta_s - \theta_r}{[1 + (\alpha \psi)^n]^m}, \tag{3}$$

$$K_w = K_s \frac{\left(1 - (\alpha \psi)^{n-1} (1 + (\alpha \psi)^n)^{-m}\right)^2}{(1 + (\alpha \psi)^n)^{\frac{m}{2}}}, \tag{4}$$

where $\theta_r$ is the residual water content, $\theta_s$ is saturated water content (equal to porosity), $\alpha$, n and m = 1-(1/n) are empirical parameters, and $K_s$ is the saturated hydraulic conductivity. The heat flow equation follows Fourier's law and accounts for conduction and convection of heat (Appendix A). The differential equations are solved with a finite difference method (Euler





integration), which requires a soil profile with a discrete number of layers having homogenous soil properties (Jansson and Karlberg, 2011).

In the following, all other main processes represented in the CoupModel set-up are described. For a detailed description of the associated mathematical expressions, see Appendix A. At the lower boundary, water may leave the soil column by deep percolation. In the present study, two different lower boundary conditions were applied: The first boundary condition assumes a non-saturated soil profile. If a pre-defined pressure head is surpassed in the lowest layer, outflow occurs as a function of the hydraulic conductivity (*free drainage*), whereas no flow occurs if the pressure head is below the specified limit. The second

boundary condition may represent saturated conditions and a variable ground water table. Here, outflow is calculated with a seepage equation dependent on the depth and spacing distance to a drain. (Jansson and Karlberg, 2011)

Infiltration capacity governs the infiltration of water at the upper boundary, and it is a function of top-most layer's saturated hydraulic conductivity and the pressure gradient to the surface. If the infiltration rate is exceeded by the water available for infiltration or if over-saturation leads to an upward movement of the soil water, water may runoff laterally. Water loss by

evapotranspiration consists of bare soil evaporation and vegetation transpiration, which in the present study was a mowed lawn. The individual evapotranspiration components were calculated using the Penman-Monteith equation (Monteith, 1965), which is mainly governed by aerodynamic and surface resistances (evaporation) as well as stomatal resistance (transpiration). The potential transpiration is limited by the availability of soil water within the rooting depth of the plants and reduced by low ground temperatures. Finally, a snow cover may be built-up based on the air temperature at times of precipitation. Snow melt

and refreezing was calculated with an empirical function depending on air temperature, global radiation and surface heat flow. (Jansson and Karlberg, 2011)

**2.3 Model set-up and parametrization**

Soil moisture was simulated at 133 sites in Switzerland where meteorological data was available from an on-site or nearby meteorological station and each site was parametrized as a grassland location (*all sites*, Fig. 1a, Table 1). At a subset of 35

sites (*monitoring sites*), in situ soil moisture measurements were available, which were used for benchmarking the statistical landslide forecast model (see section "Soil moisture data"). At a subset of 14 selected sites (*reference sites*) in situ soil moisture measurements were used to assess the soil moisture simulations from the CoupModel. They were selected because they were located on the same plot as the meteorological station and below grassland vegetation, i.e. the soil moisture sites which were disregarded for model assessment were located at far distance from the meteorological site (> 2 km) and/or located in a forest

and thus not representative for the grassland parametrization.

The goal of this study was to define a common parametrization set for all sites, hence no site-specific calibration was conducted. Most parameter values were left at the default values of the CoupModel (documented in detail by Jansson and Karlberg, 2011) whereas others were (i) adjusted to fit observed soil moisture dynamics, (ii) they were taken from literature values or (iii) they were defined by author's judgment. A description of key parameters and chosen values are given in

Appendix A. The plant properties for the grassland cover were defined by literature values (leaf area index = 0.6 $m^2$ $m^{-2}$,





canopy height = 0.25 m, maximum root depth = 0.6 m) and they were defined as homogenous throughout the season. The values for the maximal conductance of fully open stomata ($g_{max}$ = 0.03 m s$^{-1}$) and the critical pressure head for reduction of potential water uptake ($\psi_c$ = 1500 cm water) were chosen by comparison with observed soil moisture variation. Seasonal snow cover dynamics were compared with snow depth measurements available at some of the sites and tuned by adjusting the

empirical snow melt coefficients ($m_T$ = 1.5 kg °C$^{-1}$ m$^{-2}$ d$^{-1}$, $m_f$ = 0.1 m$^{-1}$, $m_{Rmin}$ = 1.5e-8 kg J$^{-1}$). Different lower boundary conditions were tested including both saturated and unsaturated conditions, and best parameter values ($\psi_{Max}$ = 10 cm for unsaturated conditions; $z_{p2}$ = 7.5 m, $d_{p2}$ = 100 m for saturated conditions) were defined by comparison with observed soil moisture variation.

Soil profiles were defined by 11 model layers of increasing thickness with depth and a total thickness of 300 cm. To reflect

regional geological differences, soil hydraulic properties were varied for each site. Since no laboratory or field data was available to measure or fit the soil hydraulic parameters ($\alpha$, n, $\theta_r$, $\theta_s$, $K_s$), they were predicted from easier available soil texture and bulk density values using the Rosetta3 H3w pedotransfer function (Zhang and Schaap, 2017). Rosetta3 was derived from a dataset containing 2134 soil samples from North America and Europe (Schaap et al., 2001). The underlying dataset also includes data from Switzerland giving confidence that the pedotransfer function is suited for the application to soils in

Switzerland. The required soil texture and bulk density values were derived from two sources: (1) At the 35 monitoring sites, texture and bulk density measurements were available from soil samples taken at various depths along the soil profiles where soil moisture sensors were installed (referred to as *soil samples*). The data was provided by the operators of the soil moisture monitoring sites (see section "Soil moisture data"). (2) At all 133 sites, texture and bulk density estimates were extracted from the SoilGrids system (referred to as *SoilGrids*) which provides global predictions of various soil properties based on machine

learning techniques (Hengl et al., 2017). SoilGrids is available in 250 m resolution at seven standard depths (0, 5, 15, 30, 60, 100 and 200 cm) and permits estimates of texture and bulk density on a global scale. Comparison of the texture split values between the soil samples and SoilGrids datasets for the 35 common monitoring sites and two depth bands (0–0.4 m, 0.4–1.0 m) reveals a narrower value distribution of SoilGrids with particularly coarse fractions missing (Fig. 2a, e).

To further test the model sensitivity to the soil hydraulic properties, four soil profiles with uniform texture (referred to as

*uniform-texture profiles*) were defined and soil hydraulic properties were defined based on literature values. The profiles include homogeneous parameter values at all depths and correspond to extreme and typical coarse- and fine-grained soils. If the derived soil hydrological properties are compared between all sources, a narrower value distribution is again visible for the SoilGrids dataset compared to the soil samples, however, median values are similar or in the same order of magnitude (Fig. 2b-d, f-h). Parameter values of the four uniform-texture profiles vary considerably, whereas the normal fine-grained uniform-

texture profile shows similar parameter values as the median values derived from the soil samples and SoilGrids.

**2.4 Meteorological input data**

The CoupModel was run at hourly time-steps using measurements of the five properties precipitation, air temperature, wind speed, relative humidity and global radiation. Data were available from meteorological stations of various monitoring



networks: (1) SwissMetNet is the automatic monitoring network of the national meteorological agency MeteoSwiss. Data from

114 monitoring sites were used in this study dating back up to 1981. (2) DTN (former MeteoGroup) is a provider of meteorological measurements and a partner network of MeteoSwiss, and two sites were included in this study (https://www.dtn.com). (3) The Long-Term Forest Ecosystem Research Programme (LWF) of the Swiss Federal Research Institute WSL conducts research on forest ecosystem processes on forested monitoring plots in Switzerland and Europe. At 14 sites, from which 8 were included in this study, collocated meteorological measurements are taken in an open-field in less than

2 km distance from the plots (Rebetez et al., 2018). (4) The Swiss FluxNet initiative includes 8 long-term ecosystem sites with eddy-covariance flux measurements in Switzerland (www.swissfluxnet.ch). In this study, meteorological measurements from two sites were included. (5) Finally, meteorological measurements from one site at the Rietholzbach Research Catchment were included, which is operated by the Land-Climate Dynamics Group (ETH, Zurich; https://iac.ethz.ch/group/land-climate-dynamics/research/rietholzbach.html).

At each site, all available meteorological data was included from the first point at which all five parameters were available (as early as 1981) until the end of 2019. Data gaps are generally short (hours to days) and were linearly interpolated in the CoupModel except for precipitation, for which zero precipitation was assumed. Each complete time series was replicated prior to the first measurement by two randomly selected consecutive hydrological years (spin-up period). Both data gaps and spin-up periods as well as the first 3 months after the spin-up period were removed for the statistical analysis.

**2.5 Soil moisture data**

For assessing the CoupModel performance and for comparison of the simulation-based forecast model with a forecast model based on measurements (see section "Statistical landslide forecast model"), soil moisture measurements from 35 sites in Switzerland were included in the study (*monitoring sites*, Fig. 1a, Table 1). Soil moisture is measured with TDR or capacitance-based sensors at various depths along a soil profile with the lowest sensors typically located at depths of 80–120 cm. The

dataset includes sites from monitoring networks of various research institutions and authorities, and measurements were available earliest from 2008 until end of 2018. The dataset was compiled and described in detail in a previous study (Wicki et al., 2020).

**2.6 Landslide data**

Landslide records from the Swiss flood and landslide damage database (Swiss Federal Research Institute WSL, Hilker et al.,

2009) were used to fit the landslide forecast model (see section "Statistical landslide forecast model"). The database includes landslide events which were identified from news articles in all of Switzerland since 1972. Records include coordinates, the date and time of the event (if known), and an event description.

For this study, events recorded from 1981 until end of 2019 were included. Following the approach of Wicki et al. (2020), deep-seated and human-induced landslides (e.g. pipe breaks, road embankment slips) were removed if explicitly mentioned in

the event description. Further, if no time of occurrence was specified, it was set to 12:00 p.m., or, if the approximate timing





was given in the event description, the timing was assumed (e.g. 09:00 a.m. for "in the morning"). In total, 2969 events were included in this study (Fig. 1a), 1041 of which contained a precise time information.

## 2.7 Statistical landslide forecast model

To assess the information content of the simulated soil moisture dynamics for regional landslide warning, a statistical
framework was applied to the simulated soil moisture time series. This framework was developed in a previous study where it was applied to in situ measured soil moisture in Switzerland (for a detailed description see Wicki et al., 2020). It included first a normalization of soil moisture values by the minimum and the 99.5 percentile values to represent soil saturation, and the calculation of mean and standard deviation saturation at each soil profile for all model layers until 140 cm depth. At each profile, periods of continuous saturation increase (*infiltration events*) were then identified automatically based on the mean
saturation time series. Each infiltration event was characterized by a set of event properties derived from both the mean and standard deviation time series (see Table 2). Finally, infiltration events were flagged as *landslide triggering* or *landslide non-triggering* providing that a landslide was observed or not observed, respectively, during the event period and within a specific distance from the modelling site (*forecast distance*).

A multiple logistic regression model was then fitted to the set of infiltration events where the binary outcome variable (i.e. the
landslide triggering class "yes" or "no") was modelled as a function of the independent infiltration event properties (explanatory variables). The logistic regression model yields a probability for each infiltration event to belong to the landslide triggering class (*triggering probability*). A 5-fold cross-validation (CV) scheme was applied to assess the robustness of the model fit with equally sized folds and randomly selected infiltration events. Four folds were used to fit the model and the remaining fold was used as the to make predictions. This approach is referred to as the *validation set approach*, as opposed to
the *all dataset approach* where the statistical model is fit to all the infiltration events.

## 2.8 Skill of the landslide forecast

To assess the forecast goodness of each specific statistical model fit, receiver operating characteristic analysis (ROC) was performed according to Fawcett (2006). First, a threshold was applied to reclassify the triggering probabilities of the infiltration events into the binary triggering classes *landslide triggering* or *landslide non-triggering*. A confusion matrix was constructed
between observed and modelled triggering classes counting the number of true positives (TP), true negatives (TN), false positives (FP) and false negatives (FN). The true positive rate, $TPR = TP / (TP + FN)$, and false positive rate, $FPR = FP / (FP + TN)$ were computed accordingly. To assess the overall potential of a model fit for multiple thresholds, the threshold was varied 5000 times in equal steps between the minimum and maximum triggering probability, thus resulting in 5000 confusion matrices. The 5000 TPR and FPR pairs were then plotted in a 2D plot (*ROC space*), resulting in a cumulative curve (*ROC*
*curve*) for which the area under the curve (*AUC*) was computed.

The forecast goodness of different model fits was assessed qualitatively by comparing the ROC curve and quantitatively by comparing the AUC value, which corresponds to the probability of listing a positive instance higher than a negative instance





if sorted by the observed triggering class. A perfect classifier plots near the top left corner of the ROC space (AUC = 1.0) whereas it is no better than random guessing if it plots along the (0/0) to (1/1) diagonal (AUC = 0.5). To assess the distance-

dependence of the forecast models, each model set-up was fit using eight different forecast distances ranging in equal steps from 5 to 40 km. We chose the ROC curve and AUC value as performance indicators because they assess the general forecast goodness of a statistical model in contrast to many other performance indicators that quantify the forecast goodness of specific threshold values (Piciullo et al., 2020).

## 3 Results

### 3.1 General model performance


The performance of the soil moisture model and the corresponding triggering probabilities according to the landslide forecast model are illustrated for a sample period from mid-April to mid-May 2015 (Fig. 3a). During this time period, a series of intense precipitation events led to wide-spread landslide activity in central Switzerland with numerous landslides observed from 30 April until 4 May 2015 (black dots on Fig. 1b and colour filled background on Fig. 3a, b). Profile saturation for a subset of 11

sites in the region of interest (coloured points on Fig. 1b) remained low and inhomogeneous prior to the landslide period. It increased to near-saturated conditions and remained very wet for a couple of days which coincides with the period of landslide activity. This development is confirmed by the landslide forecast model, which shows a low triggering probability at the beginning of the period (red horizontal lines; note that landslide probability is only computed for periods of saturation increase). Triggering probability increased significantly across all sites during the period of landslide activity and descended again after

that.

These patterns can be compared to in situ soil moisture measurements at the same sites and the corresponding landslide forecasts of a forecast model fitted to the soil moisture measurements (Fig. 3b). Temporal evolution of the profile saturation shows similar regional-scale patterns with variably-saturated conditions during the first half of the sample period followed by an increased saturation during the period of landslide activity. Further, the measurement-based landslide forecast model shows

a similar triggering probability development as the simulation-based model with significantly higher triggering probabilities for all sites during the days of observed landslide events compared to the periods prior and after that. Yet, distinct differences are apparent: The temporal evolution of simulated profile saturation appears to be more homogeneous between different sites, the desaturation immediately after an infiltration event is slower and it reaches drier conditions after sustained periods of no infiltration. Triggering probabilities are generally lower for the measurement-based landslide forecast model.

**3.2 Performance assessment of the soil hydrological model**

The agreement between simulated and observed soil wetness was analysed for the 14 reference sites by the mean error (ME) and coefficient of determination ($R^2$) statistics computed for the hourly soil moisture values. The skill of the model set-up was generally solid but strongly varied from site to site and with depth of sensors (Fig. 4a). Best agreement was found for the top-





most sensors (median ME = 0.00 m³ m⁻³, median R² = 0.55–0.60). At depths of 30 and 50 cm, ME values were similar but R²
values were lowest across all depths (median R² = 0.40–0.45). R² statistics were better at 80 cm depth (median R² = 0.50–
0.55), however, mean error was greater than at all other depths (median ME = 0.02–0.05 m³ m⁻³), indicating too dry conditions
at the lower boundary probably due to overestimation of deep percolation. This skill is comparable or slightly lower than
reported skills for other soil moisture models used in landslide early warning (e.g. Brocca et al., 2008; Thomas et al., 2018) or
for CoupModel set-ups with different purposes (e.g. Conrad and Fohrer, 2009; He et al., 2016). However, it has to be noted
that these models are mostly validated for one or two sites only and were partially calibrated site-specifically.

Not much difference in model skill was found between using a lower boundary condition without groundwater (Fig. 4a) and
with groundwater (Fig. 4b). When a lower boundary with groundwater was defined, ME statistics remained very similar
(median ME = 0.00–0.05 m³ m⁻³) and R² statistics slightly improved at the lowest depths (median R² = 0.45 at 50 cm, median
R² = 0.60 at 80 cm). Best model fit of the groundwater-based model set-up was found for a parametrization indicative for a
well-drained site.

Another important part in the parametrization is the site-specific definition of the soil hydrological properties. Since texture
and bulk density information from soil samples were available for the monitoring sites only, they were derived from a gridded
product (SoilGrids) in addition in order to be able to apply the CoupModel with the same general set-up at all sites. Comparison
of a soil samples based model setup (Fig. 4a, b) with a model set-up based on SoilGrids information (Fig. 4c, d) revealed very
similar model skill, with a slightly decreased mean error (median ME = 0.00 m³ m⁻³ at all depths) and slightly larger range of
R² values for the SoilGrids-derived model set-up. This indicates that SoilGrids adequately represents the regional variation in
soil hydrological variability and can be used to extend the model to all other sites. Further, the effect of having no regional
variation in soil hydrological properties was tested by deriving them from the normal fine-grained uniform-texture profile (Fig.
4g). Mean error statistics remained in a similar range (median ME = 0 m³ m⁻³ at all depths), however R² values were
significantly lower at all depths (median R² = 0.30–0.55). This demonstrates the value of including regionally-varying soil
hydrological properties.

Finally, large sensitivity of the model skill was found for variation of the saturated hydraulic conductivity, which was tested
by deriving the soil hydrological properties from the other, more extreme uniform-texture profiles. Above-average $K_s$ values
were defined for profiles representing extreme coarse-grained and normal coarse-grained soils (Fig. 4e, f), and $K_s$ values were
below average for the extreme fine-grained uniform texture-profile (Fig. 4h). Model skill showed a very poor model fit for the
coarse-grained profiles (median R² = 0.05–0.20) and very high mean error values indicative for too dry conditions (median
ME = 0.25 m³ m⁻³ at all depths). Model fit was better for the extreme fine-grained profile (median R² = 0.40–0.50) but ME
statistics showed too wet conditions (median ME = -0.10 m³ m⁻³ at all depths). This indicates that the SoilGrids and soil samples
derived saturated hydraulic conductivity values are in an adequate order of magnitude.

One important result of our soil moisture model assessment was the fact that the deviation between model and measurement,
i.e. the *residuals,* were not varying randomly, but had a seasonal trend (Fig. 5a, b, residuals were computed as mean daily
values across all 14 sites). With a CoupModel set-up using SoilGrids information and a bottom boundary condition with





groundwater, winter months showed positive anomalies (i.e. modelled soil moisture was drier than observed) whereas negative anomalies (i.e. wetter than observed) were apparent during summer months. Both effects were more pronounced in near-surface layers. Further, near-surface layers showed wetter than observed anomalies after the exceptionally dry summer in 2015 indicating that the extreme drying could not be reproduced very well. We explain this underestimation of the seasonal variation with an underestimation of evapotranspiration in summer (too wet conditions in summer when evapotranspiration is high) and a generally faster drainage than observed (too dry conditions in winter when evapotranspiration is low). Comparison with the long-term evolution of simulated soil moisture (mean daily values across all 14 sites, Fig. 5c) showed no apparent trend, thus the intra-year variability of residuals can be explained with variations in precipitation and evaporation.

### 3.3 Performance of the statistical landslide forecast model

### 3.3.1 Simulated versus observed soil moisture; 35 monitoring sites

ROC curves and AUC values for a CoupModel set-up with groundwater and using soil hydrological properties derived from soil samples are shown in Fig. 6a. For comparison with a measurement-based statistical model fit, the dataset contains the 35 monitoring sites only and modelling periods were limited to the same periods as soil moisture measurements were available (2008–2018). ROC curves of all forecast distances clearly deviated from the (0/0) to (1/1) line and most AUC values were larger than 0.8 indicating that all forecast distances bore some information content on the regional landslide activity. Forecast goodness was strongly distance dependent with short forecast distances having a better forecast goodness (AUC = 0.86 at 5 km, AUC = 0.79 at 40 km, all dataset approach). This is in good agreement to the results of Wicki et al. (2020) for measured soil moisture. The robustness of the statistical model fit was assessed by comparison with the AUC values and ROC curves of the validation set approach (Fig. 6e). Values were very similar for most forecast distances indicating a robust model fit, however, robustness was slightly lower at short forecast distances probably due to the low number of landslides records (7% of all landslides were within the 5 km radius of the 35 sites; see Table 3).

Compared to a statistical model derived from measured soil moisture (Fig. 6d, h), the number of infiltration events was similar, yet the overall forecast goodness of the measurement-based forecast model was lower at all forecast distances (AUC = 0.83 at 5 km, AUC = 0.72 at 40 km, all dataset approach). This is remarkable as the simulated soil moisture was shown to contain specific uncertainty particularly related to the long-term water storage in the soil. We explain the better forecast goodness of the simulation-based landslide forecast model by a more homogeneous representation of infiltration characteristics in space (less influence of local conditions such as groundwater influence or preferential infiltration) and in time (no drift or trend as might be observed for some erroneous or badly coupled soil moisture sensors), as well as a more homogeneous site representation (number of sensors and depth levels included in the analysis).



### 3.3.2 Simulated soil moisture using in situ soil physical properties versus using SoilGrids

A similar forecast goodness resulted for a simulation with SoilGrids-derived soil hydrological properties compared to a simulation with soil hydrological properties derived from soil samples (Fig. 6b, f). AUC values and number of infiltration

events were in the same range (AUC = 0.87 at 5 km, AUC = 0.78 at 40 km, all dataset approach), and ROC curves followed a similar shape with more robust model fits at large forecast distances. This finding is in line with the similar goodness of fit as shown in the previous section and demonstrates the validity of using soil hydrological properties derived from SoilGrids. It permits to extend the approach to all 133 sites for most of which no in situ soil sample information was available.

### 3.3.3 Increase of number of soil moisture sites

Extending the analysis to all 133 sites and to the entire input data time period (1981–2019) resulted in a considerably higher number of infiltration events (N = 142'311) and thus much smoother ROC curves (Fig. 6c, g). Further, the model fits became very robust even at short forecast distances (i.e. same AUC values for the all dataset and validation set approach). AUC values were slightly lower than when the 35 monitoring sites were used, but in the same range (AUC = 0.87 at 5 km, AUC = 0.76 at 40 km), and ROC curves bulged slightly less to the top, indicating a worse performance for optimistic thresholds.

Increasing the number of sites also increased the area and number of landslides covered, as illustrated with Table 3. When all 133 sites were used, almost the whole country and all landslides were covered by using a 15 km forecast distance. When the 35 monitoring sites were used only (as was the case for the measurement-based forecast model), the same coverage is only possible when a 40 km forecast distance is used. This is due to the lower number of sites and because the available sites are distributed inhomogeneously including large gaps in alpine areas and in the eastern part of the country (Fig. 1a).

### 370 3.3.4 Sensitivity of the landslide forecast model to the definition of the lower boundary condition and soil properties

The sensitivity of the landslide forecast model to changes in the lower boundary condition was assessed by testing different lower boundary parametrizations for CoupModel set-up using all 133 sites (Fig. 7, grey boxes highlight the model parametrization that was chosen for the goodness of fit analysis). Low sensitivity of the landslide forecast goodness was found for variations of the lower boundary conditions without groundwater (Fig. 7a), which was defined by the maximum pressure

head of the lowest layer above which outflow occurs as gravitational outflow. In contrast, when the lower boundary was defined with a seepage function, the landslide forecast goodness was very variable. Best results were obtained for a fairly steep gradient to the drain, i.e. a larger depth to drain (Fig. 7b) or a shorter distance to drain (Fig. 7c). This indicates a better landslide forecast goodness for well drained sites. As shown previously, the landslide forecast goodness was similar for both, a CoupModel parametrization with and without groundwater, which is in line with the very similar goodness of fit values for

the two parametrizations.

Low sensitivity of the landslide forecast model was found when using soil hydrological properties derived from uniform-texture profiles (Fig. 8), resulting even in a slight forecast goodness increase for the extreme and normal coarse-grained





uniform-texture profiles. This is surprising, since by using uniform-texture profiles the regional variation of soil hydrological properties is disregarded and $K_s$ values partially deviate substantially from what can be expected in reality, both of which was

reflected with a substantially worse agreement with measured soil moisture in a previous section. The reasons behind this are studied in more detailed in the discussion section.

### 3.4 Most important explanatory variables for landslide forecast model

In the previous section, the landslide prediction models were fitted including all explanatory variables (also referred to as *event properties*) as listed in Table 2. In order to analyse the contribution of individual explanatory variables to the overall forecast

goodness, the landslide prediction model was fitted to individual explanatory variables only, as illustrated in Fig. 9 (first column) where AUC values were plotted for different statistical model fits. Explanatory variables can be grouped into variables describing the antecedent wetness conditions (underlain in red) and into variables describing the infiltration event dynamics (underlain in orange). For reference, a model fit including two explanatory variables only (*antecedent saturation* and *saturation change*, second column) and a model fit including all explanatory variables (third column) were plotted too. As to be expected,

the forecast goodness of individual explanatory variables was significantly lower than when all explanatory variables were included. Further, model fits using the two explanatory variables antecedent saturation and saturation change had almost the same forecast goodness as if all event properties were used.

When looking at individual explanatory variables in detail, distinct differences become apparent between statistical model fits based on simulated and measured soil moisture. For the simulation-based landslide forecast models, the increase of the forecast

goodness was mostly driven by explanatory variables that describe the triggering event dynamics (e.g. saturation change during the infiltration event, maximum 3-hours infiltration rate, infiltration rate, standard deviation change, Fig. 9a, d, g). Inversely, for a measurement-based landslide forecast model, explanatory variables related to the antecedent wetness conditions were more important (e.g. antecedent saturation, 2-week preceding maximum saturation, Fig. 9k).

The worse performance of explanatory variables related to the antecedent wetness conditions for the simulation-based forecast

models can be related to the reduced ability of the CoupModel set-up to reflect long-term seasonal water storage, as described previously (Fig. 5). The better forecast goodness of explanatory variables related to the triggering event dynamics of the simulation-based landslide forecast model can be explained by a more homogeneous site set-up, no impact by very site-specific conditions (e.g. preferential flow, interaction with a local groundwater table, interaction with the vegetation), and by the elimination of measurement errors (e.g. sensor drift, sensor uncertainties, bad sensor contact to surrounding).

The better performance of explanatory variables related to event dynamics compared to those related to antecedent conditions was even more accentuated in the case of the extreme coarse-grained uniform-texture profile with a better forecast goodness of most of the event dynamics-related explanatory variables (Fig. 9g). Conversely, the antecedent saturation explanatory variable even showed a slight forecast goodness decrease.



## 4 Discussion

### 4.1 Limitations of the soil moisture model

The soil moisture model incorporates errors and uncertainties connected to the parametrization and the quality of the input data, limiting the availability to reproduce soil moisture variation as observed with soil moisture sensors. A large component of uncertainty originates from the definition of the soil hydrological properties, which in previous studies were shown to have a great impact on simulated soil moisture and landslide forecasts (e.g. Thomas et al., 2018). Here, uncertainty is added both from the definition of the site-specific texture and bulk density values as well as from the estimation of the soil physical properties with a pedotransfer function. No substantial differences in the goodness of fit of simulated versus observed soil moisture were found between using soil hydrological properties derived from soil samples and taken from SoilGrids. Yet, a decrease in coefficient of correlation was found when using the same normal fine-grained uniform-texture profile for all sites. From that we can conclude that the soil hydrological properties differences between using soil samples and SoilGrids weigh smaller than the missing regionalization inferred by using a uniform-texture profile only. This underlines the importance of using regionally varying soil physical information for simulating soil moisture which is often omitted due to a lack of field data or because too many parameters may lead to overfitting problems (e.g. Posner and Georgakakos, 2015; Zhao et al., 2019b). Larger uncertainty is probably introduced by the use of a pedotransfer function to infer the soil hydrological properties from soil physical information. This point cannot be validated directly since field data on site-specific soil hydrological properties is missing, however, the large ME spread across the 14 reference sites points towards partially incorrect residual $\theta_r$ and saturated water content $\theta_s$ values. Further, many studies highlight that pedotransfer functions incorporate a bias towards loamy agricultural soils and lack a representation of soil structure such as the presence of macropores or concretisations (e.g. Or, 2020; Zhang and Schaap, 2019). This may lead to an underestimation of $K_s$ values which in return impacts surface runoff generation, water infiltration and discharge (Fatichi et al., 2020).

A second major source of uncertainty originates from the definition of homogeneous upper and lower boundary conditions. In general, seasonal soil moisture variation was underestimated, a problem reported also in other modelling studies (Okkonen et al., 2017; Orland et al., 2020; Zhuo et al., 2019a), and which may be partially attributed to the definition of the vegetation and soil resistances and the potential evapotranspiration calculation. Calibration is difficult due to missing evapotranspiration measurements. We compared our evapotranspiration estimates with a countrywide evapotranspiration estimation function for grassland locations depending on elevation (Hydrological Atlas of Switzerland HADES, Menzel et al., 1999) and with estimations from lysimeter measurements at the Rietholzbach site (RHB, Hirschi et al., 2017). It was shown that evapotranspiration estimates slightly underestimated the HADES values, however they followed the same elevation dependence (Fig. 10a). When comparing with field lysimeter data, evapotranspiration estimates were lower too (Fig. 10b), but followed the general seasonal variation and showed similar inter-annual variation except for the year of 2008 (Fig. 10c). This may explain the underestimated drying-out of the model compared with the observations as shown previously, which could be improved by a more elaborate or site-specific vegetation parametrization. Nevertheless, it has to be noted that the





evapotranspiration data presented here is only weakly representative as it is based on simulations as well and shows regional values (in case of HADES) and lysimeter measurements were available for one site only.

At the lower boundary of the soil profile, the definition of well-drained conditions showed best results. However, soil
hydrological conditions might differ substantially for individual sites if shallow groundwater tables are present (Marino et al., 2020) or if soil depths vary between the sites (Anagnostopoulos et al., 2015), hence a site-specific parametrization might improve the goodness of fit with observed soil moisture variation. While no seepage data on regional scales was available for calibration or validation, site-specific definition of lower boundary conditions could be achieved by consideration of nearby groundwater level measurements or regional groundwater distribution maps when defining the depth and distance to drain for
a lower boundary with groundwater.

Finally, when comparing the goodness of fit with observed soil moisture measurements, it has to be noted that the soil moisture measurements bear uncertainties too and might be erroneous or contain a signal shift or trend due to bad contact with the surrounding material, sensor deterioration, or structural changes in soil. Thus, a thorough quality assessment is needed when using soil moisture data for calibration or validation. Further to that, measurement uncertainties of the meteorological input
data may have a considerable impact if comparing the simulated soil moisture time series with observed soil moisture variation. Particularly rainfall data may lack from undercatch problems pronounced at high and exposed locations, which was not corrected for in the meteorological time series used in this study (MeteoSwiss 2020, personal communication).

Following from this, the goodness of fit might improve significantly by applying a site-specific calibration scheme. While this would allow to reflect complex local conditions, site-specific calibration is limited by the availability of field data for
calibration (e.g. soil moisture or evapotranspiration data) and it is restricted by the number of parameters to fit. Further, site-specific calibration is not possible if the model should be applied at places where no measurements are available (e.g. for complementing an existing soil moisture monitoring network). Grouping sites into areas of similar physiographic characteristics, e.g. based on soil type, land-use or geological data, to further constrain parameter values may be a first step towards this (Fatichi et al., 2016). Finally, data assimilation techniques, often applied with land-surface models (Reichle et al.,
2014) or in models used for landslide early warning (Krøgli et al., 2018), could help to adjust for the seasonal misfit of the long-term water storage term, but again depend on the availability of field data and are thus limited to locations with soil moisture measurements.

### 4.2 The value of simulated soil moisture for landslide early warning

Our results showed that the simulation-based landslide forecast models performed slightly better than a forecast model based
on soil moisture measurements, implying that simulation-based soil moisture information is overall more representative for regional landslide occurrence. This can be explained by considering different time scales and the hydrological processes associated with them: The overall improvement with a simulation-based forecast model is based on a better representation of the triggering conditions, notably the infiltration of water during precipitation or snow melt events. In this domain, processes typically range in time scales of hours to days and are highly influenced by local factors such as preferential infiltration along



macropores and fissures in the ground, surface ponding and runoff due to an impeding surface layer, interception by the vegetation cover, interactions with impeding layers within the soil columns or at the soil-bedrock transition, as well as interactions with a ground-water table. While the spatial variability of these processes can be high in reality, they are simplified and represented homogeneously in the model. In addition to this homogenization in the process domain, the statistical variation is homogenized over time (no sensor errors or drifts as may be observed for single sensors) and between sites (depth levels

and number of depth levels considered). We assume that the homogenized representation of the processes in space leads to a more robust statistical model fit and hence an improved landslide forecast goodness.

With regard to the antecedent conditions, the measurement-based landslide forecast model performed better. The hydrological processes associated with this domain are governed by the redistribution of soil water after rainfall events, by the steady drainage of water at the bottom of the profile and by evapotranspiration from the soil surface. Simplification and

misrepresentation of some of these processes in the CoupModel set-up may lead to an underestimation of the seasonal soil moisture variation, which in Switzerland is high, with generally wet conditions from fall to spring and a dry period with intermitted wetting events during summer (Pellet and Hauck, 2017). A limited seasonal representation may reduce the forecast model's ability to separate triggering from non-triggering conditions, as reported for regional landslide forecast models where regions with different seasonal soil moisture variation were compared (Thomas et al., 2020). Particularly in regions with

a high seasonal evapotranspiration variation, wet and dry periods may be controlled by different soil moisture fluxes with vertical fluxes being dominant during dry periods and lateral fluxes during wet periods (e.g. Grayson et al., 1997), thus a physically-based representation of the processes is important and a spatial representation may improve the seasonal soil moisture variation.

While the two process domains (antecedent conditions and event dynamics) can be analysed individually, they also influence

each other due to the limited value distribution of soil saturation ranging from 0 % (residual soil water content) to 100 % (full saturation). If water drains quicker, more pore space is available for rainfall to infiltrate in the next event and intense rainfall events may show a stronger soil moisture response. Conversely, soil moisture responses to precipitation events are weaker in wetter and more fine-grained soils due to slower infiltration, less available pore space due to pre-saturation and more surface runoff due to impeding conditions near the surface. Hence, in a more conductive soil, the statistical model is more able to

separate triggering from non-triggering events at the expense of the loss of long-term water storage information. These effects were most clearly visible when soil hydraulic properties of an extremely coarse-grained uniform-texture profiles were used in the CoupModel, which showed an even better landslide forecast goodness. The fast drainage causes the evapotranspiration loss to be ineffective and thus, the model becomes more a representation of rainfall characteristics demonstrating the high information content in precipitation for landslide prediction.

To validate this hypothesis, we applied the same statistical model to the precipitation time series only, which were used to drive the soil moisture model. Individual precipitation events were defined as continuous periods of rainfall (> 0.5 mm h$^{-1}$) separated by gaps of at least 3 hours of no rainfall. Precipitation event sums were computed and (1) normalized with the total porosity of the uppermost 100 cm taken from SoilGrids and resulting event sums were normalised with the 99.5 percentile of





each time series to represent some soil information (Fig. 11a), or (2) event sums were solely normalized by 99.5 percentile of
the event time series (Fig. 11b). While the first statistical model includes some information on the regional soil physical
conditions (porosity from SoilGrids), the latter includes rainfall information only. The number of precipitation events was
about the double compared to the simulations with soil hydraulic properties (i.e. not all precipitation events were manifested
as infiltration events in simulations of soil water dynamics). Despite the larger number of precipitation events (the classification
between triggering and non-triggering events is typically easier with a higher number of events), the distance-averaged AUC
value dropped from 0.82 for simulations of soil water dynamics to 0.79 for rainfall signatures. But both approaches (based on
precipitation and infiltration events, respectively) showed similar landslide forecast goodness and similar forecast distance
dependence, highlighting that the landslide forecast goodness is mainly driven by spatial rainfall variation.

While it is discouraging that similar landslide forecast goodness can be achieved with a forecast model that is based on rainfall
information only or with a heavily simplified model representation (e.g. based on the extreme coarse-grained uniform-texture
profiles), the benefit of a well parametrized physically-based soil moisture transfer model or of using soil moisture
measurements remains in the quantification of the antecedent wetness conditions, particularly if a strong seasonal variation
persists. This is often missed by less physically-based approaches using e.g. antecedent wetness indexes or antecedent
precipitation indexes (e.g. Brocca et al., 2012).

## 5 Conclusions

The present analysis demonstrated a high information content of simulated soil moisture for regional landslide activity, which
was even higher than when in situ soil moisture measurements were used. The forecast goodness of such a landslide warning
system strongly depends on the distance between soil moisture stations and landslide location, i.e. on the soil moisture station
density, because of more robust model fits at near forecast distances and a greater spatial coverage of landslide events and
regions of interest. The advantage of soil moisture simulations over in situ soil moisture measurements is the better
representation of triggering event conditions, probably due to homogenization of the hydrological processes and the site
representation (number and depths of sensors included). On the other hand, the simulation-based forecast model performed
worse than the measurement-based at reproducing critical antecedent saturation conditions, possibly due to the inadequate
representation of the long-term water storage.

In comparison with a statistical landslide forecast model that only uses precipitation or that simulates soil moisture with very
simplified (uniform) soil hydraulic properties, the main added value of a comprehensive physically-based soil moisture
simulation is the representation of critical antecedent wetness conditions. To improve the soil moisture model in this respect,
further explorations in the use of site- or regional-specific calibration schemes are needed and other calibration data than soil
moisture measurements should be incorporated.





**Figures**

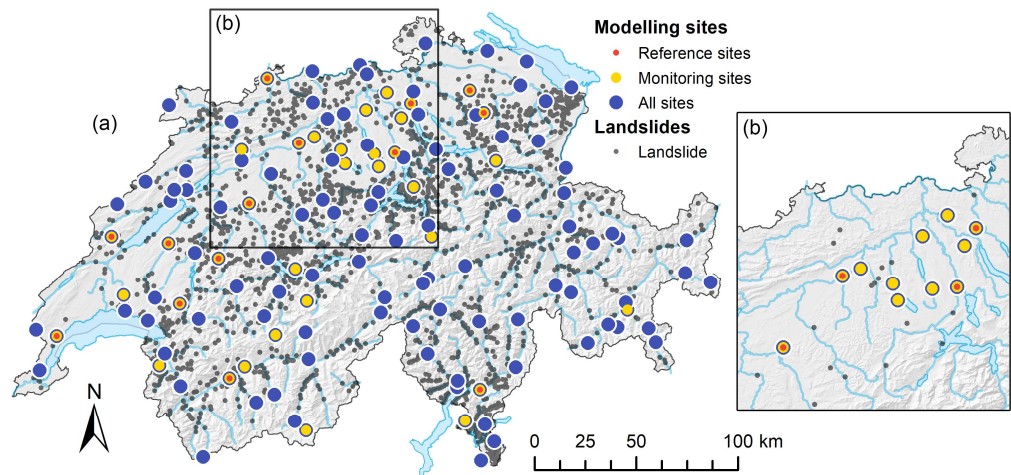


**Figure 1: Map of Switzerland showing the locations of the soil moisture modelling sites (coloured points) and the landslide locations (black points) including (a) all sites and all landslides of the entire study period from 1981 to 2019 and (b) a subset of sites and landslides that were triggered during a series of rainfall events between 30 April and 3 May 2005.**



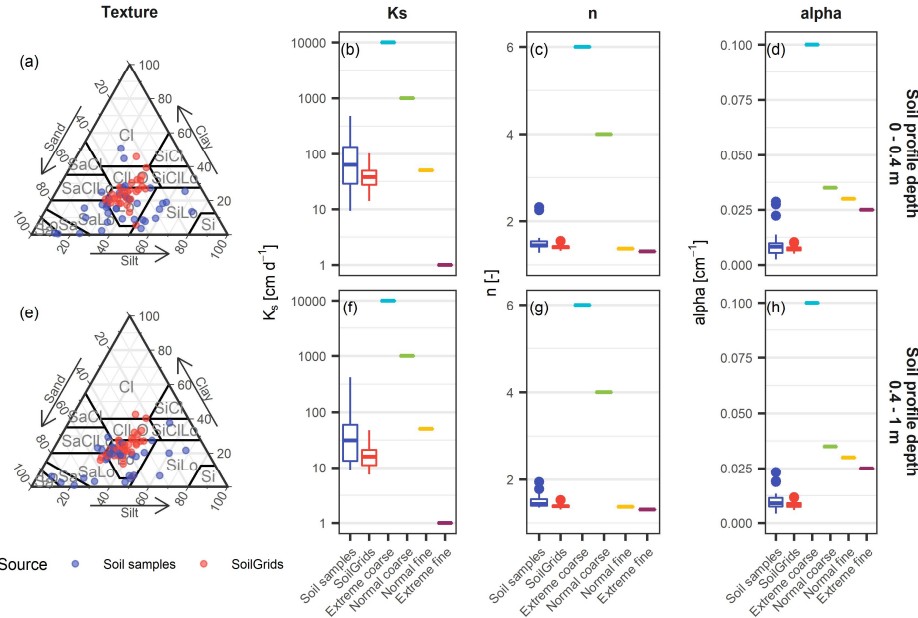

**Figure 2: Soil texture splits (a, e) and soil hydraulic properties $K_s$ (saturated hydraulic conductivity), $n$ (van Genuchten coefficient) and $\alpha$ (van Genuchten coefficient) (b-d, f-h) of the 35 monitoring sites averaged for the model layers in 0–40 cm (a-d) and 40–100 cm depth (e-h); the point and boxplot colors indicate different sources of information (soil samples, SoilGrids, uniform-texture profiles).**


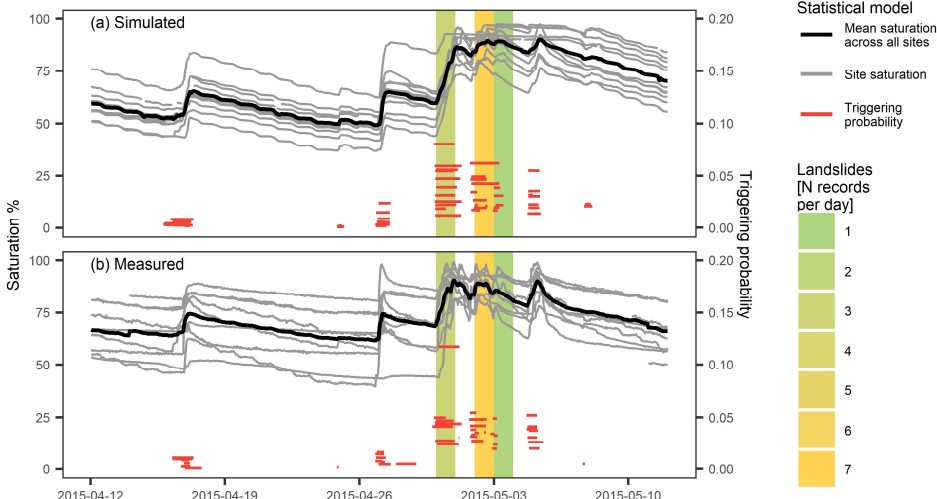

Figure 3: Profile saturation (grey lines) of a selection of 11 sites in central Switzerland (as depicted on Figure 1b) with the mean profile saturation across all selected sites (black line) during a period of increased landslide activity in April and May 2015 for (a) simulated and (b) measured soil moisture. Colour filled background denotes days with observed landslide events within 15 km of any of the sites with the colour indicating the number of landslide records. The red lines show the associated landslide triggering probability from the statistical model (based on the nine infiltration event properties listed in Table 2) at each site which was computed for periods of saturation increase only.

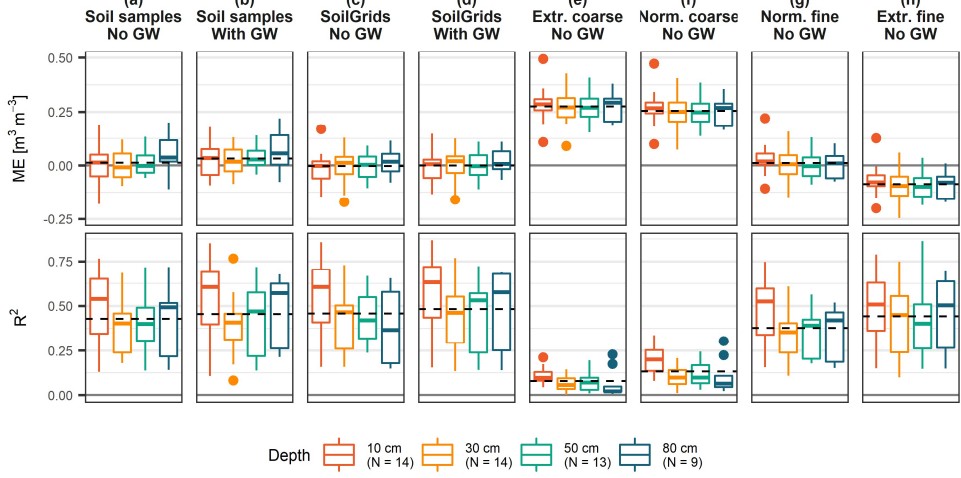

Figure 4: Goodness of fit of simulated versus measured soil moisture variation at the 14 reference sites: Mean error (ME, top panels) and coefficient of determination ($R^2$, bottom panels) by sensor depths (different colours) for various CoupModel parametrizations (a-h). Lower boundary conditions with and without groundwater (GW) are distinguished.

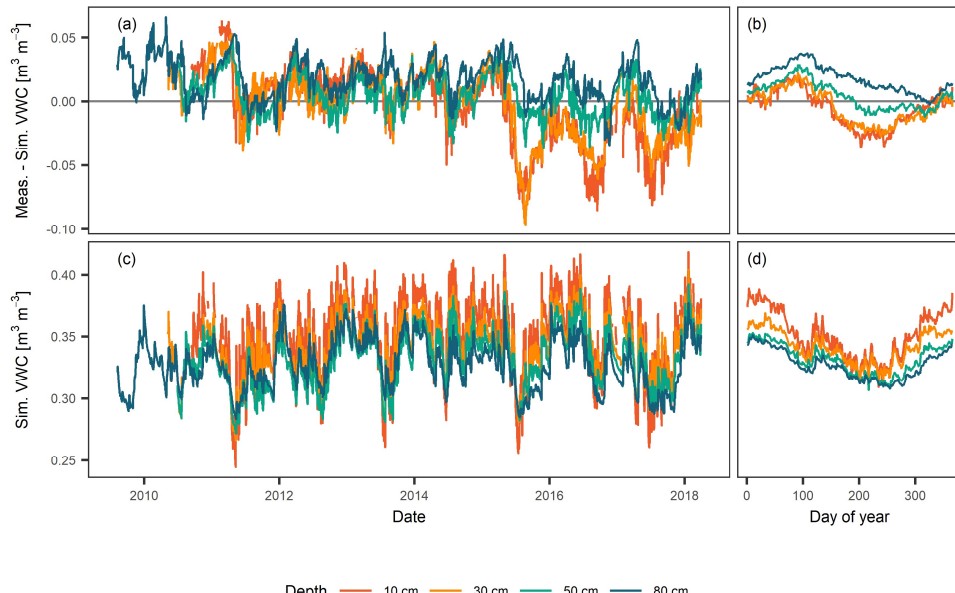


**Figure 5: Temporal evolution and seasonal variation of mean daily residual VWC (a, b), i.e. deviation between simulated and observed soil water content, and mean daily simulated VWC (c, d) across all 14 reference sites by sensor depths (different colours) for a CoupModel set-up using soil hydrological properties derived from SoilGrids and a lower boundary condition with groundwater.**





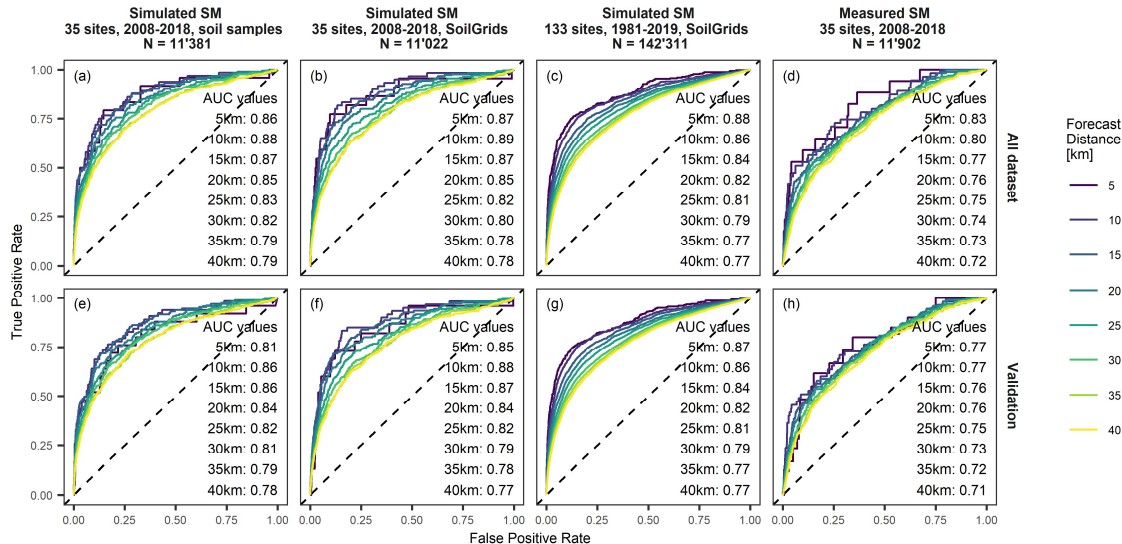


**Figure 6: ROC plots and AUC values of landslide forecast model fits based on simulated soil moisture (SM) at the 35 monitoring sites including soil hydrological properties from soil samples (a, e) and SoilGrids (b, f), at all 133 sites including soil hydrological properties from SoilGrids (c, g) and based on measured soil moisture at the 35 monitoring sites (d, h). All CoupModel set-ups include a lower boundary condition with groundwater. Upper panels (a-d) show landslide forecast model fits using all the dataset whereas lower panels (e-h) show model fits based on the 5-fold cross-validation scheme.**


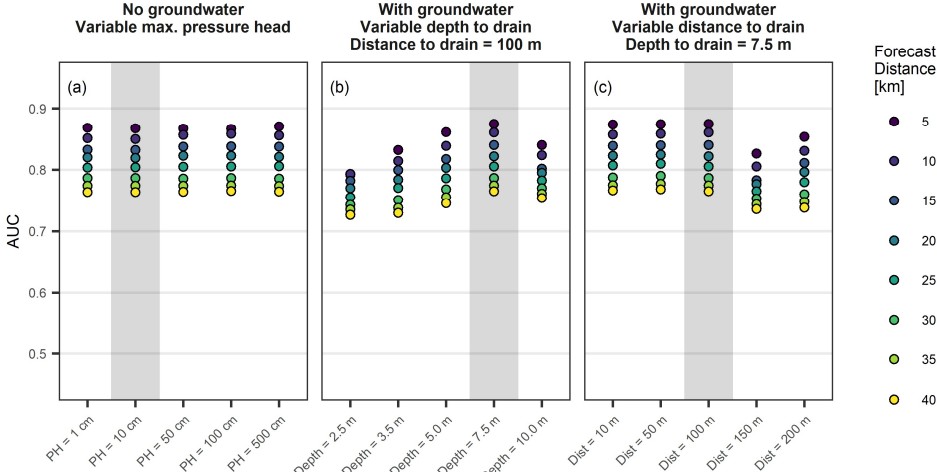

**Figure 7: AUC values of landslide forecast model fits with different parametrizations of the lower boundary condition by varying (a) the maximum pressure head of the lowest layer above which exfiltration occurs, (b) depth to the drain, and (c) the distance to the drain. Grey shaded model runs correspond to the CoupModel parametrization used in all other analyses.**





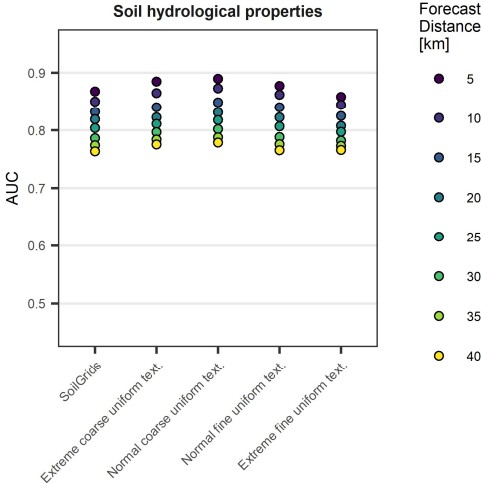


**Figure 8: AUC values of landslide forecast model fits based on CoupModel set-ups with varying soil hydrological properties (derived from SoilGrids and uniform-texture profiles) and a lower boundary condition with groundwater.**

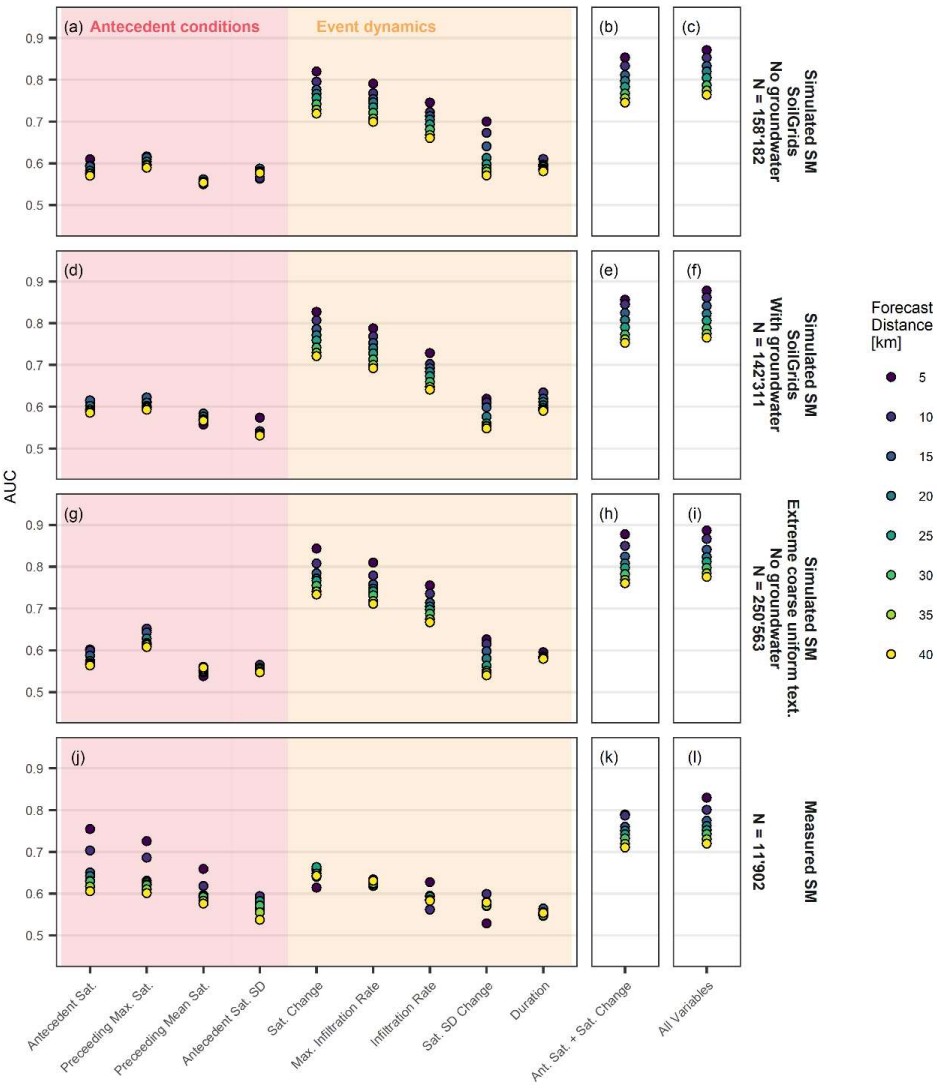

**Figure 9:** AUC values of model fits based on simulated soil moisture (SM) at all 133 sites (upper three rows) and based on measured soil moisture at the 35 monitoring sites (lowest row). Model fits include individual explanatory variables only (a, d, g, j), the explanatory variables antecedent saturation and saturation change only (b, e, h, k) and all nine explanatory variables (c, f, i, l).





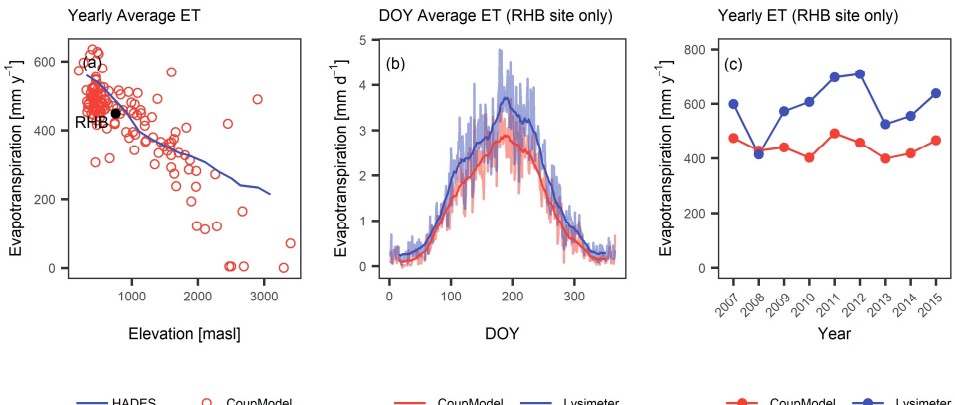

**Figure 10: Simulated evapotranspiration (ET) from the CoupModel versus validation data. (a) Yearly average ET versus elevation function from the Swiss Hydrological Atlas, (b) day of the year average ET and (c) yearly ET at the Rietholzbach site (RHB) vs. lysimeter data measured at this site.**


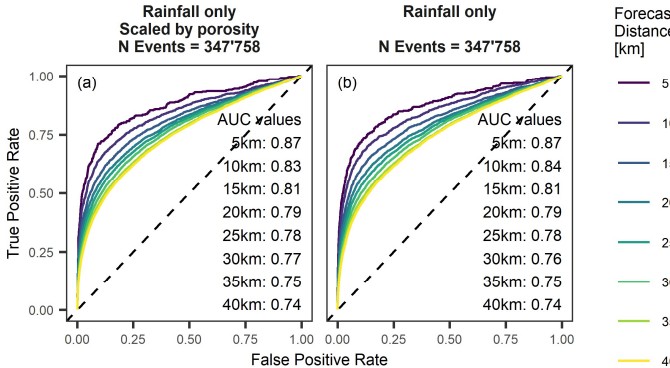

**Figure 11: ROC curves and AUC values of model fits (a) based on normalized rainfall information scaled by porosity and (b) based on normalized rainfall information only for all 133 sites and the entire modelling period (1981–2019).**





**Tables**

**Table 1: Sets of sites, available datasets and number of sites.**

| Sites set | Texture and bulk density information | | Co-located soil moisture measurements | N sites |
|---|---|---|---|---|
| | **SoilGrids** | **Soil samples** | | |
| All sites | Yes | No | No | 133 |
| ∟ Monitoring sites | Yes | Yes | Yes | 35 |
| ∟ Reference sites | Yes | Yes | Yes (grassland only, at < 2 km distance) | 14 |

**Table 2: List of event properties to describe infiltration events. To classify between triggering and non-triggering infiltration events,**
**the nine event properties marked with "x" are used. Time series of the mean of water saturation and standard deviation (SD) of**
**saturation were used.**

| Process domain | Event property description | Name | Profile Mean | Profile SD |
|---|---|---|---|---|
| Antecedent conditions | Saturation at the onset of infiltration event | Antecedent sat. | x | x |
| | 2-week preceding maximum saturation | 2 week-prec. max sat. | x | |
| | 2-week preceding mean saturation | 2 week-prec. mean sat. | x | |
| Event dynamics | Saturation change during an infiltration event | Sat. change | x | x |
| | Infiltration rate | Infiltration rate | x | |
| | Maximum 3-hours infiltration rate | Max inf. rate | x | |
| | Event duration | Duration | x | |

**Table 3: Percentage of country (area of Switzerland) and number of landslides (percentage of all landslides recorded from 1981 to**
**2019) covered by the soil moisture simulations and measurements as a function of the forecast distance used.**

| Forecast distance | All sites (133 sites) | | Monitoring sites (35 sites) | |
|---|---|---|---|---|
| | **% of total area** | **% of all landslides** | **% of total area** | **% of all landslides** |
| 5 km | 22.6 | 26.8 | 6.4 | 7.1 |
| 10 km | 65.6 | 73.7 | 22.1 | 26.0 |
| 15 km | 91.4 | 95.4 | 41.5 | 49.6 |
| 20 km | 98.6 | 99.2 | 58 | 65.8 |
| 25 km | 99.7 | 99.8 | 70.3 | 76.4 |
| 30 km | 100.0 | 100.0 | 79.3 | 83.1 |
| 35 km | 100.0 | 100.0 | 87.0 | 88.6 |
| 40 km | 100.0 | 100.0 | 92.6 | 93.7 |





## Appendix A

Table A1: Key equations of the CoupModel used for this study (Jansson and Karlberg, 2011 for more details).

| Nr | Equation | Description |
|---|---|---|
| **Deep percolation (unsaturated conditions)** | | |
| (A1) | $q_{deep} = \begin{cases} k_{wlow}, & \psi > \psi_{Max} \\ 0, & \psi \leq \psi_{Max} \end{cases}$ | Deep percolation, $q_{deep}$, under unsaturated conditions as a function of the hydraulic conductivity of the lowest layer, $k_{wlow}$, and the simulated pressure head of the lowest layer, $\psi$. Deep percolation occurs if the maximum pressure head, $\psi_{Max}$, is exceeded. Below the threshold, no flow of water occurs. |
| **Deep percolation (saturated conditions)** | | |
| (A2) | $q_{deep} = \dfrac{8k_{slow}(z_{sat} - z_{p2})^2}{d_{p2}^2}$ | Saturated deep percolation ($q_{deep}$) depends on the saturated hydraulic conductivity of the lowest layer ($k_{slow}$). Drainage is at a spacing distance $d_{p2}$ and at depth $z_{p2}$, both of which are parameters. The simulated ground water table depth is at $z_{sat}$. |
| **Infiltration and surface runoff** | | |
| (A3) | $q_{in} = \begin{cases} q_{th}, & i_{cap} > q_{th} \\ i_{cap}, & i_{cap} \leq q_{th} \end{cases}$ | The infiltration rate ($q_{in}$) is simulated as a function of the surface infiltration capacity ($i_{cap}$). It equals the precipitation throughfall rate ($q_{th}$) if throughfall is smaller than the infiltration rate. |
| (A4) | $q_{surf} = \begin{cases} a_{surf}(W_{pool} - w_{pmax}), & W_{pool} > w_{pmax} \\ 0, & W_{pool} \leq w_{pmax} \end{cases}$ | Surface runoff ($q_{surf}$) is generated if throughfall exceeds the infiltration capacity and a surface pool of water is formed, with $W_{pool}$ being the total water amount. The amount of water which can be stored ($w_{pmax}$) is a parameter and $a_{surf}$ is an empirical coefficient. |
| **Potential transpiration** | | |
| (A5) | $L_v E_{tp} = \dfrac{\Delta R_n + \rho_a c_p \dfrac{(e_s - e_a)}{r_a}}{\Delta + \gamma\left(1 + \dfrac{r_s}{r_a}\right)}$ | The Penman's combination equation (Monteith, 1965) is used to calculate potential transpiration ($E_{tp}$). It depends on net radiation ($R_n$), the difference of saturation and actual vapour pressure ($e_s-e_a$), the aerodynamic resistance ($r_a$) and the surface resistance ($r_s$). It further depends on air density ($\rho_a$), specific heat of air ($c_p$), latent heat of vaporization ($L_v$) and the psychometric constant ($\gamma$) which are all considered physical constants, and the slope of saturated vapour pressure vs. the temperature curve ($\Delta$). |
| (A6) | $r_a = \dfrac{ln^2\left(\dfrac{z_{ref} - d}{z_0}\right)}{k^2 u}$ | The aerodynamic resistance ($r_a$) is depending on the wind speed (u) measured at the reference height ($z_{ref}$). It is proportional to the displacement height (d) and inversely proportional to the roughness length ($z_0$). k is the von Karman's constant. |
| (A7) | $r_s = \dfrac{1}{\max(A_l g_l, 0.001)}$ | Surface resistance ($r_s$) inversely proportional to the leaf area index ($A_l$) and the leaf conductance ($g_l$). |
| (A8) | $g_l = \dfrac{R_{is}}{R_{is} + g_{ris}} \dfrac{g_{max}}{1 + \dfrac{(e_s - e_a)}{g_{vpd}}}$ | The leaf conductance ($g_l$) is calculated by the Lohammar equation (Lindroth, 1985; Lohammar et al., 1980). It depends on global radiation ($R_{is}$) and the vapour pressure deficit ($e_s-e_a$) with $g_{ris}$, $g_{max}$ and $g_{vpd}$ being parameter values. |





**Actual transpiration**

(A9)
$$E_{ta} = E_{ta}{}^* + f_{umov}\left(E_{tp}{}^* - E_{ta}{}^*\right)$$

Actual transpiration ($E_{ta}$) may compensate for soil layers with water stress by a two-step calculation. The left term ($E_{ta}{}^*$) corresponds to the water uptake without compensation. The right term is the difference of the potential transpiration ($E_{tp}{}^*$, with a reduction due to interception evapotranspiration) and actual transpiration, and the degree of compensation is governed by the parameter $f_{umov}$.

(A10)
$$E_{ta}{}^* = E_{tp}{}^* \int_{z_r}^{0} f\left(\psi(z)\right) f\left(T(z)\right) r(z)$$

Response functions for soil water potential, $f(\psi(z))$, and for soil temperature, $f(T(z))$ are used to reduce potential transpiration ($E_{tp}{}^*$) to calculate actual transpiration ($E_{ta}{}^*$). It is calculated for each soil layer and integrated with $r(z)$ being the distribution of relative root density and $z_r$ being the maximal root depth.

(A11)
$$f\left(\psi(z)\right) = min\left(\left(\frac{\psi_c}{\psi(z)}\right)^{p_1 E_{tp}+p_2}, f_\theta\right)$$

Transpiration is reduced under dry conditions by stomatal mechanism and xylary tissue resistance and becomes zero at the wilting point becomes. $p_1$, $p_2$ and $\psi_c$ are parameters, and $f_\theta$ is an additional response function (not shown).

(A12)
$$\int_{z_r}^{z} r(z) = \frac{1 - e^{-k_{rr}(z/z_r)}}{(1 - r_{frac})}$$

The distribution of root density is represented in exponential form. Below a depth z, the fraction of roots depends on the extinction coefficient $k_{rr}$ whereas $r_{frac}$ is a parameter. The integral calculated on the entire soil profile equals unity.

**Soil evaporation**

(A13)
$$L_v E_s = \frac{\Delta(R_{ns} - q_h) + \rho_a c_p \frac{(e_s - e_a)}{r_{as}}}{\Delta + \gamma \left(1 + \frac{r_{ss}}{r_{as}}\right)}$$

The Penman's equation (Monteith 1965) is used for calculation of soil evaporation ($E_s$). It is calculated from the surface latent heat flux ($L_v E_s$) which depends on the energy available at the surface ($R_{ns}-q_h$, i.e. available net radiation minus soil surface heat flux from previous step), the aerodynamic resistance ($r_{as}$), the surface resistance ($r_{ss}$), the difference of saturation and actual vapour pressure ($e_s-e_a$). All other terms are equal to the terms in (A5).

(A14)
$$r_{ss} = \max\left(0, r_{\psi 1} max\left(\psi_s - r_{\psi 2}, 0\right) - r_{\psi 3}\delta_{surf}\right)$$

Soil surface resistance ($r_{ss}$) is governed by the parameters $r_{\psi 1}$, $r_{\psi 2}$ and $r_{\psi 3}$. It accounts for the water tension in the uppermost layer ($\psi_s$) and the mass balance at the soil surface ($\delta_{surf}$).

**Radiation processes**

(A15)
$$R_{n,tot} = R_{is}(1 - a_r) + R_{lnet}$$

Net radiation ($R_{n,tot}$) is the sum of global radiation ($R_{is}$) minus the surface albedo ($a_r$), and net long-wave radiation ($R_{lnet}$).

(A16)
$$R_{lnet} = 86400\sigma(\varepsilon_s(T_s + 273.15)^4 - \varepsilon_a(T_a + 273.15)^4)$$

The Brunt's formula is used to calculate net long-wave radiation ($R_{lnet}$) where the surface emissivity ($\varepsilon_s$) is assumed equal to 1 and the atmosphere emissivity ($\varepsilon_a$) is calculated by Konzelmann et al. (1994). $T_s$ is the surface temperature.

**Snow dynamics**

(A17)
$$P_{rain} = P\left(1 - Q_p\right)$$

The fraction of solid precipitation ($Q_p$) determines the snow and rain partitioning of precipitation (P).





(A18)
$$Q_p = \begin{cases} min\left(1, (1 - f_{liqmax}) + f_{liqmax}\dfrac{T_a - T_{RainL}}{T_{SnowL} - T_{RainL}}\right), & T_a \leq T_{RainL} \\ 0, & T_a > T_{RainL} \end{cases}$$

$Q_p$ is a function of air temperature ($T_a$), with $T_{RainL}$ and $T_{SnowL}$ being parameters describing the temperature range of mixed ice and liquid water precipitation and $f_{liqmax}$ being the maximal liquid water content of falling snow (equals 0.5).

(A19)
$$M = M_T T_a + M_R R_{is} + \frac{f_{qh} q_h(0)}{L_f}$$

Snow melt (M) is calculated from a temperature function ($M_T$) and air temperature ($T_a$), a solar radiation function ($M_R$) and global radiation ($R_{is}$), as well as from surface heat flow ($q_h$), a scaling coefficient ($f_{qh}$) and the latent heat of freezing ($L_f$).

(A20)
$$M_T = \begin{cases} m_T, & T_a \geq 0 \\ \dfrac{m_T}{\Delta z_{snow} m_f}, & T_a < 0 \end{cases}$$

Snow melt and refreezing are governed by the empirical parameters $m_T$ and $m_f$. Refreezing is simulated only for a limited surface layer and is thus inversely proportional to snow depth ($\Delta z_{snow}$).

**Soil heat flow**

(A21)
$$q_h = -k_h \frac{\partial T}{\partial z} + C_w T q_w$$

Soil heat flow ($q_h$) is calculated as the sum of conduction (first term) and convection (second term) where $k_h$ is the soil heat conductivity, T is temperature, $C_w$ is heat capacity of liquid water and $q_w$ is the liquid water flow. In this model set-up, latent heat flow by water vapour was disregarded.

(A22)
$$\frac{\partial(CT)}{\partial t} - L_f \rho \frac{\partial \theta_i}{\partial t} = \frac{\partial}{\partial z}(-q_h)$$

The heat flow equation includes changes in sensible and latent heat contents (left side) and input or output of heat from the soil layer (right side) and is calculated for each soil layer. It follows from combining (A36) with the law of energy conservation. C is the heat capacity, T is temperature, $L_f$ is latent heat of freezing, $\rho$ is density, $\theta_i$ is the water content of ice.

**Table A2: Description of the most important parameter values used in the CoupModel set-up along with the associated equations (Jansson and Karlberg, 2011 for more details).**

| Symbol | Description | Unit | Value | Eq. |
|---|---|---|---|---|
| **Deep percolation, unsaturated conditions** | | | | |
| $\psi_{Max}$ | Maximum pressure head in lowest layer, above which outflow occurs | cm | 10 | (A1) |
| **Deep percolation, saturated conditions** | | | | |
| $z_{p2}$ | Drain level depth | m | 7.5 | (A2) |
| $d_{p2}$ | Spacing distance to drain | m | 100 | (A2) |
| **Infiltration and surface runoff** | | | | |
| $a_{surf}$ | Empirical coefficient used to calculate runoff from surface pool. | 1 d$^{-1}$ | 0.8 | (A4) |
| $w_{pmax}$ | Maximum water amount stored in surface pool. | mm | 0 | (A4) |
| **Potential transpiration** | | | | |
| $z_0$ | Roughness length | - | 0.1 | (A6) |
| $d$ | Displacement height | - | 0.66 | (A6) |





| | | | | |
|---|---|---|---|---|
| $z_{ref}$ | Height above ground of wind speed, air humidity and air temperature measurements. | m | 2 | (A6) |
| $A_l$ | Leaf area index | m² m⁻² | 0.6 | (A7) |
| $g_{ris}$ | Global radiation intensity at which light response is at half-light saturation. | J m⁻² d⁻¹ | 5e+6 | (A8) |
| $g_{max}$ | Maximum conductance of fully open stomata. | m s⁻¹ | 0.03 | (A8) |
| $g_{vpd}$ | Vapour pressure deficit at which stomatal conductance is reduced by 50%. | Pa | 100 | (A8) |
| **Actual transpiration** | | | | |
| $f_{umov}$ | Degree of compensation for compensatory water uptake. | - | 0.6 | (A10) |
| $z_r$ | Maximum rooting depth | m | -0.6 | (A10) |
| $p_1$ | Empirical coefficient for soil water potential response function. | 1 d⁻¹ | 0.3 | (A11) |
| $p_2$ | Empirical coefficient for soil water potential response function. | kg m⁻² d⁻¹ | 0.1 | (A11) |
| $\psi_c$ | Pressure head above which potential water uptake is reduced. | cm water | 1500 | (A11) |
| $r_{frac}$ | Fraction of roots remaining below a given root depth. | - | 0.1 | (A12) |
| **Soil evaporation** | | | | |
| $r_{\psi 1}$ | Governing parameter for the calculation of the surface resistance. | s m⁻¹ | 0.5 | (A14) |
| $r_{\psi 2}$ | Governing parameter for the calculation of the surface resistance. | s m⁻¹ | 300 | (A14) |
| $r_{\psi 3}$ | Governing parameter for the calculation of the surface resistance. | s m⁻¹ mm⁻¹ | 80 | (A14) |
| **Snow dynamics** | | | | |
| $T_{RainL}$ | Temperature above which all precipitation is rain. | °C | 2 | (A18) |
| $T_{SnowL}$ | Temperature below which all precipitation is snow | °C | 0 | (A18) |
| $f_{qh}$ | Contribution coefficient of ground heat flow on snow melt. | - | 0.5 | (A19) |
| $m_T$ | Temperature coefficient for snow melt calculation. | kg °C⁻¹ m⁻² d⁻¹ | 1.5 | (A20) |
| $m_f$ | Efficiency constant for refreezing calculation. | m⁻¹ | 0.1 | (A20) |

**Data and code availability**

The meteorological, soil moisture and landslide data were kindly provided by the cited institutions; the CoupModel can be downloaded from https://www.coupmodel.com/.

**Author contribution**

AW carried out the soil moisture simulations and statistical analysis, and wrote the paper. PEJ provided guidance for the soil moisture simulations. PL contributed to the preparation of the soil hydrological properties. CH and MS supervised the work.

All co-authors provided guidance on the paper's research and contributed to the paper.



**Competing Interests**

The authors declare that they have no conflict of interest.

**Acknowledgments**

This research project was financially supported by the Swiss National Science Foundation (project number 175785) and is part

of the programme Climate Change Impacts on Alpine Mass Movements of the Swiss Federal Research Institute WSL. We thank MeteoSwiss, Joachim Schung (DTN), Matthias Häni and Peter Waldner (LWF), Sabina Keller (Swiss Flux-Net) and Sonia I. Seneveriatne and Martin Hirschi (Land-Climate Dynamics Group of ETH Zurich) for providing the meteorological input data, as well as soil moisture data for this study.

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
