# Peer review of "Simulated or measured soil moisture: Which one is adding more value to regional landslide early warning?"

_Hydrology and Earth System Sciences, 2021_

## Referee Comment (RC2)

[referee-annotated manuscript omitted]

---

## Author Response (AR1)

**Response to reviewer 1**

In the manuscript the authors compare modeled and measured soil moisture in the context of landslide prediction in Switzerland. Measurements are available at 35 sites across Switzerland, while the 1D soil moisture transfer model CoupModel providing estimates of soil moisture is set up at 133 sites. The landslide prediction model used is based on different soil moisture metrics and optimized by using ROC-statistics.

The authors carry out different comparisons/experiments: comparing measured and modeled soil moisture over 14 reference sites, comparing performances of the landslide prediction model based on measured or modeled soil moisture, studying the impact of boundary conditions of the hydrological model, and the impact of soil parametrization (using soil samples, uniform texture sets, or soil properties obtained with pedotransfer functions and the SoilGrid database).

The authors find that modeled soil moisture is outperforming measured moisture, that the model is sensitive to boundary conditions, and performances worsen as the distance between soil moisture and landslide locations increases.

The manuscript is clear, well written and structured, and the topic addressed is relevant. Overall, the manuscript is worth publication in hess.

We thank the reviewer for the generally positive response and the constructive comments which we addressed below.

Nevertheless, there is one important aspect which is worth addressing and discussing more in details, concerning the choice and results of the hydrological model. It has been shown in previous studies that vertical flow is the dominant process leading to landsliding, compared to lateral flow (e.g., Iverson 2000), but lateral flow becomes essential for adequate description of initial soil moisture conditions (e.g., Mirus et al., 2017, Leonarduzzi et al., 2020). This idea and the fact that the model seem to reproduce mostly just event dynamics is confirmed by several of the results:

- Figure 4: the better R2 for shallowest depth (typically better matching the patterns of meteorological forcing)
- Underestimation of seasonal variations of soil moisture
- Line 382-383: "resulting even in a slight forecast goodness increase for extreme and normal coarse-grained uniform-texture profiles". Using a highly conductive soil which drains quickly, basically reduce the model to "get rid" of initial conditions and just represent the current infiltration event (i.e., your estimate of soil moisture is basically matching P-ET). But these soils are the one giving the worst match to measured soil moisture
- 9: modeled moisture outperforming measured moisture for event dynamics but not for antecedent condition metrics.

All these aspects, lead me to think that what is happening is that the hydrological model is actually just using the information in the recent meteorology (transformed into saturation estimate using soil parameters), while the "memory" component of saturation is not well represented/useful for landslide prediction. This is sort of the opposite of what one would expect in terms of information context in a "antecedent condition" metric, as typically saturation is considered a "cause", while recent rainfall the "trigger". The authors checked this by comparing the results to a simple rainfall-based prediction and indeed find similar performances. It would be worth exploring, or at least addressing, if using a combination of measured soil moisture (providing antecedent conditions) and rainfall event dynamics (possibly accounting also for soil properties), would actually lead to better landslide predictions (e.g., logistic regression using antecedent conditions saturation metrics

measured/observed and rainfall event metrics). This would not invalidate the work presented or any of the findings but would definitely be a more complete/objective answer to the question the authors pose in the title: "Simulated or measured soil moisture: Which one is adding more value to regional landslide early warning?".

We acknowledge that the model is better at reproducing the event conditions and worse at characterizing initial saturated conditions. We addressed this in the discussion section and attributed the underrepresentation of the seasonal soil moisture cycle mainly to the definition of a common parametrization of the upper and lower boundary conditions [lines 456-476 in the revised manuscript]. We used a common parametrization set in order to be able to apply the model at sites where no site-specific calibration is possible (due to missing soil moisture measurements, soil moisture measured at distance from the meteorological site or located in the forest), which often is the case with landslide early warning systems where soil moisture is simulated. We agree that the use of a one-dimensional model which does not account for lateral water transport may add to the missing seasonal cycle too. We chose a 1D modelling approach due to computational restraints while still permitting a good representation of infiltration characteristics. We will discuss the motivation behind the common parametrization and the model choice in more detail in the revised paper. [lines 90-91, 100-102, 170-173 in the revised manuscript]

As the reviewer suggested, we fitted a landslide forecast model using antecedent saturation and event rainfall amount only using the same logistic regression model. We used both measured and simulated soil moisture and we compared these model fits to a landslide forecast model using rainfall information only (Figure S1). The analysis was conducted using all common sites and the same time periods (35 monitoring sites, 2008 – 2018), hence the lower number of infiltration events. Compared to using rainfall only, the forecast goodness increases at most forecast distances if antecedent soil moisture is used too, except for the 5 km forecast distance. This may be due to a low robustness of the statistical model fit at short forecast distances due to a low number of landslide triggering events. The forecast goodness improvement is more pronounced if measured soil moisture is used and it is only marginal if using simulated soil moisture, which is in line with the above discussion about the underrepresentation of the seasonal water storage. We will include Figure S1 in the revised paper and expand the discussion. [lines 551-557 in the revised manuscript]

As mentioned in the reviewer's comment, in an applied context, soil moisture information will be used primarily as an antecedent metric to complement rainfall information. In that sense, the characterization of the antecedent saturation should be improved which could be achieved by the use of site-specific or regional-specific parametrization, or the use of more spatially integrated models. The objective of this paper, however, was not to produce specific soil moisture thresholds or an operationally applicable model, but rather to test the overall information content of soil moisture information and to highlight the differences between using simulated or measured soil moisture. We will clarify the objectives of this study in the revised paper. [lines 90-91 in the revised manuscript]

[Figure]

**Figure S1** ROC curves and AUC values of model fits (a) using rainfall amounts only, (b) using antecedent saturation (measured and rainfall amounts) and (c) using antecedent saturation (simulated) and rainfall amounts, for all 35 monitoring sites for the period of 2008 to 2018.

Finally, some minor comments:

- In Figure 5, are the lines showing the average across 14 sites for each depth?
  Yes, the lines show the average profile saturation across 14 sites, with the profile saturation being the mean saturation of all model depths or sensor depths. We will make this clearer in the revised manuscript. [line 595-598 in the revised manuscript]
- It could be interesting (although probably worth including only in the appendix), to see the trends of soil moisture observed and measured at different depths.
  We have added the measured soil moisture time series to the plot (Figure S2). Further, we have added a trend line to all time series. While a negative trend is visible at the measured time series, there is no apparent trend for the modelled time series which might be due to an underrepresentation of evapotranspiration in the model. However, the negative trend of the measurements might as well be influenced by non-stationary soil moisture measurement time series, e.g. due to compaction of the soil or enhanced root development around the sensors, as some of the sensors have been running for almost 10 years. Further, different numbers of sensors were integrated over different time periods as not all sensors have been installed at the same time which adds to the uncertainty of interpreting a long-term trend in this integrated signal. To assess the reasons for the trends in the measured time series, they could be compared to other long-term hydrological measurements such as ground-water or runoff time series from nearby stations, which would be needed if a site-specific calibration was conducted. In the revised paper, we will include the new figure and discuss the trend lines it in more detail. [lines 339-352 in the revised manuscript]

[Figure]

**Figure S2** Temporal evolution and seasonal variation of mean daily residual volumetric water content (VWC) (a, b), i.e. deviation between simulated and observed soil water content, and mean daily measured (c, d) and simulated VWC (e, f) across all 14 reference sites by sensor depths (different colours) for a CoupModel set-up using soil hydrological properties derived from SoilGrids and a lower boundary condition with groundwater. Panels c and e include trend lines by sensor depth.

Iverson, R. M. (2000). Landslide triggering by rain infiltration. Water Resources Research, 36(7), 1897–1910. https://doi.org/10.1029/2000WR900090

Leonarduzzi, E., Maxwell, R. M., Mirus, B. B., & Molnar, P. (2021). Numerical analysis of the effect of subgrid variability in a physically based hydrological model on runoff, soil moisture, and slope stability. Water Resources Research, 57, e2020WR027326. https://doi.org/10.1029/2020WR027326

Mirus, B. B., Ebel, B. A., Loague, K., & Wemple, B. C. (2007). Simulated effect of a forest road on near surface hydrologic response: redux. Earth Sugace Processes and Landforms, 32(1), 126–142. https://doi.org/10.1002/esp.1387

**Response to reviewer 2**

The paper deals with a hot topic for the researchers working on landslide hazard management, i.e. the potential improvement of (shallow) landslide predictive models at regional scale, offered by adding soil moisture information to the commonly used precipitation data. The idea of comparing the predictive performance improvement deriving from modelled or from measured soil moisture is surely of interest for some of the readership of HESS.

The paper is clearly written and organized, the data and methods are adequately described, and much valuable information is made available to scientists dealing with landslide hydrology. Overall, my judgement of the paper is positive, with just few minor points that could help further improving it (you can find them as comments in the attached annotated file).

However, I would like to share some points of discussion with the authors, leaving to them the decision on if, and to what extent, they could find some space in the revised paper.

We are happy about the overall positive judgment and appreciate the constructive comments by the reviewer that we addressed below. Further, we commented the remarks from the supplement at the end of this document.

To judge about the added value of soil moisture information for landslide prediction (and for the comparison between two sources of information, measurements and modelling), there are two tasks: assessing the long-term water balance of the slopes, which is mainly controlled by what happens at the boundaries (overland runoff generation, evapotranspiration, deep leakage at the bottom of the soil cover); simulating what happens during rapid rainfall or snowmelt infiltration events, for which boundaries are expected to be less important compared to soil hydraulic properties. The first task has to do with the antecedent conditions, which may predispose the slopes to failure; the second directly with the triggering of landslides.

The results clearly indicate that modelling the long-term processes affecting the slopes (i.e. water balance) can be quite different than modelling the dynamics of the short-term infiltration of rainfall or snowmelt. In fact, the adopted 1D physically based model satisfactorily reproduces slope response during short periods with several infiltration events (fig. 3, although the soil parameters used in these simulations are not specified), but in the long run there is some systematic mismatch between simulated and measured soil moisture (fig. 5, looking at which I would be careful with the conclusion that there is not "apparent trend" in simulated soil moisture). The authors recognize that there is an issue in the modelling of evapotranspiration, which can be underestimated of even up to 200 mm per year (fig. 10), but surprisingly, they do not elaborate on this in their attempts of simulation. Instead, all the attention is focused on the effects of changing soil hydraulic parameters and on the bottom boundary condition.

All the variables tested to improve landslide predictions (table 2) refer to the water balance of the soil cover (i.e. mean cover saturation, infiltration flux at the upper boundary, bottom boundary condition), and so it is not surprising that the shape parameters of van Genuchten retention model do not affect the predictions that much. Given the tested variables, only Ksat is important, and this is probably the reason why the coarse-grained homogeneous soil profile works well, as this profile is associated with the highest value of Ksat (and thus infiltration), so that the saturation variations are mostly sensitive to rainfall are the highest. I think this is also the reason why it does not seem that including soil moisture changes the predictive performance so much compared to using rainfall information alone: the possible effects of antecedent conditions on infiltration dynamics are lost, as the model fails reproducing the water balance, and so soil saturation trend simply follows precipitation trend.

About the bottom boundary condition, the most valuable for landslide predictions are the ones which maximize the drainage. I wonder if this is a way to compensate the underestimation of evapotranspiration: in Alpine environment (high altitude, rocky bedrock, steep slopes etc…), is it plausible that there is a groundwater table, few meters below the soil cover, affecting it? It seems to me that this bottom boundary conditions becomes more conceptual than physically based.

All in all, it is not surprising that this kind of soil moisture modelling does not add useful information for landslide prediction about antecedent conditions, being outperformed even by sparse field measurements, and that so little improvement is provided by the modelling of infiltration event dynamics (compared to precipitation alone).

Given all these considerations, some questions arise: is it worth using such a sophisticated unsaturated soil model, with so many equations and so many parameters difficult to set (table A1), when only the water balance of the soil cover is needed? Would the result of the comparison between modelled and measured soil moisture give a different result if a simpler modelling approach was used? Is it possible to conclude that the aims listed at lines 85-89 have been achieved?

Indeed, after the presented detailed study, with a rich dataset (landslides, soil properties, meteorological input, soil moisture measurements) and with a complex modelling exercise (exploring also the effects of different parametrizations), little conclusions are drawn: soil moisture measurements seem to allow a better assessment of antecedent conditions, but their use is limited by spatial resolution; soil moisture modelling requires different parametrization to provide better results. In view of this, maybe the aims of the study, and the title as well, could be reformulated in a less ambitious way.

I hope that my considerations can be of some help for the authors, for this paper or for future further elaborations of their data.

Roberto Greco

We acknowledge that the model is worse at representing the long-term water balance compared to characterizing infiltration event conditions. We discussed potential reasons for this and attributed the underrepresentation of the seasonal soil moisture cycle mainly to the definition of a common parametrization of the boundary conditions (lines 456 – 476 in the revised manuscript). The definition of common boundary conditions was needed in order to able to apply the model at locations where no site-specific calibration was possible. The motivation behind applying the model at such locations was to test the use of modeled soil moisture data to complement a soil monitoring system.

The reviewer argued that some part of this water balance misrepresentation may originate from an underestimation of evapotranspiration as some sites clearly deviate from the validation function shown in Fig. 10. It has to be noted that the validation function was developed for flat open-land grassland locations in Switzerland. In contrast, some of the meteorological sites are partially shaded due to topography or located on an oriented slope. Hence, the validation data serves only as a rough point of reference for a specific site elevation and we believe that the evapotranspiration is within reasonable ranges. We will explain the limits of the validation data in more detail in the discussion part.

Given all the simplifications (soil hydrological properties, homogeneous upper and lower boundary conditions, no lateral flow considered) we agree that a simpler model might as well produce similar results. In contrast, it would also be interesting to assess the benefit of applying a site- or regionspecific parametrization or of using a model that also considers lateral water flow. While such investigations are out of scope of this study, the dataset used here may serve as a basis for further analyses in that direction. In the revised paper, we will explicate more the choice of the model and put it into a broader context in the discussion part. [lines 90-91, 100-102, 485-496 in the revised manuscript]

Finally, the question was raised whether all the aims listed in the introduction were reached. We believe this is the case. With regards to the above discussed points, however, the aims may have been formulated too broadly. We will reformulate them and explicitly narrow them down to the use of a 1D soil hydrological model. [84-91 in the revised manuscript] In the original answer, we suggested to change the title too. Finally, we decided to leave the title as in the first submission, as we believe it still well fits to what the study covers.

Answers to comments in the supplement:

- P. 6, line 191: We will add information about the climatology in Switzerland in the revised manuscript. [lines 103-115 in the revised manuscript]
- P. 9, line 267: The simulated saturation time series were based on a Coup-Model set-up with groundwater and using soil properties from SoilGrids. We will specify this in Fig. 3 in the revised manuscript. [lines 281-283 in the revised manuscript]
- P. 9, line 274: The triggering probability remains low for all triggering events. This is due to the imbalanced dataset (very few triggering events as opposed to many non-triggering events) and commonly reported for logistic regression models for such data sets (also referred to as rare events data; King and Zeng, 2001). Nevertheless, the relative difference between triggering and non-triggering events is large enough to be detected in the ROC analysis. We will elaborate on this in the revised manuscript. [lines 291-294 in the revised manuscript]
- P. 11, line 329: We have plotted the measured soil moisture time series in as well and we have added a trend line to all plots (Figure S1). No clear trend is visible for the modelled time series, whereas a decreasing trend is apparent for the near-surface layers in the measured time series. While this might be indicative for underrepresented drying out towards the end of the study period, it might also be the result of data quality issues of the measurements resulting in reduced homogeneity of the long-term soil moisture time series which were partially running for up to 10 years (e.g. due to compaction of the soil, enhanced root development around the sensors). In a future study with site-specific calibration, this could be studied in more detail e.g. by comparing these trends with nearby long-term ground water or runoff measurements. Further to that, individual time series have different lengths and thus the depth-integrated signal shown in the plot may be influenced by partial under- or over-representation due to the simplification of the soil hydrological properties. We will elaborate both points in more detail in the revised paper and we will add the new figure. [lines 341-352 in the revised manuscript]
- We will correct the various grammatical errors highlighted.

[Figure]

**Figure S1** Temporal evolution and seasonal variation of mean daily residual VWC (a, b), i.e. deviation between simulated and observed soil water content, and mean daily measured (c, d) and simulated VWC (e, f) across all 14 reference sites by sensor depths (different colours) for a CoupModel set-up using soil hydrological properties derived from SoilGrids and a lower boundary condition with groundwater. Panels c and e include trend lines by sensor depth.

King, G. and Zeng, L.: Logistic Regression in Rare Events Data, Polit. Anal., 9(2), 137–163, doi:10.1093/oxfordjournals.pan.a004868, 2001.

**Response to reviewer 3**

The paper addresses a relevant topic. The proposed method does not have strong evidence of operational applicability at the moment, but the work represents an interesting attempt to address some issues that are being largely debated inside the community involved in landslide forecasting studies.

We thank the reviewer for the constructive feedback. We addressed the points raised in the answers below.

- English is very good.

- The state-of-the-art review and introduction are very good.

I would stress much more about the scale of the application. The Authors are attempting to propose a national-scale method. This is very ambitious and, of course, the modeling approach and the research design require some simplifications and generalizations that would not be acceptable in local-scale studies. This would be useful also to avoid some form of criticism from other reviewers and, in perspective, from readers.

Here, we tested the information content of plot-scale soil wetness data (1D soil moisture simulations) for regional landslide early warning on a national scale. The motivation behind using plot-scale data was to compare the information content of simulated soil moisture data with the information content of in situ soil moisture measurements which measure at the plot scale and for which a statistical model was developed in an earlier study. We could confirm that the plot-scale soil moisture simulations bear specific information content for regional landslide early warning and that the information content decreases with increasing distance between site and landslide, similar to the information content of soil moisture measurements. We clarified the motivation behind the use of plots-scale simulation data in the manuscript. [lines 90-91 in the revised manuscript]

- Connected with the previous point: a "test site" section is mandatory. Authors should provide a short description of Switzerland: areal extension, geology, lithology, landslide processes, climate (snow-related processes are particularly difficult to handle in landslide studies), and mainland cover types. As a reader, I would skip this part because I am familiar with the site, but I doubt all readers are as familiar as Europeans.

We have added a section "study area" that provides a physiographic description of Switzerland and that covers the points raised. [lines 103-115 in the revised manuscript]

- I have a comment on the sensitivity of results on the grain size (and, consequently, Ks values). In the case of coarse soils with high Ks, I do not think it is worth investing much effort in modeling the hydrological behavior of slopes. Originally, the empirical rainfall threshold approach was born (and ultimately found effective) just for this kind of soil and shallow landslide process. The latest efforts of the scientific community are needed because we are trying to apply the same methodology (empirical rainfall thresholds) to a wider range of settings, including finer soils and different landslide types.

We could show that if an extremely coarse-grained soil profile was used in the soil hydrological model, the information content is very similar to using rainfall information only. We agree that simulating soil moisture adds more value for finer-grained soils than to coarse-grained soils.

- The validation of landslide forecasting models with AUC may be useful to get the feeling of the sensitivity of the results. However, AUC metrics are insensitive to cut-off (threshold) values, while in

a landslide forecasting model I would expect just to find which is the most effective threshold to be used for warning.

In this study, we were not looking for a specific threshold value. Rather, we were testing different threshold values (here we varied the threshold 5000 times at equal increments) to test the overall potential or information content of soil wetness information for regional landslide early warning. Therefore, we compared different AUC values between different model formulations, which is a measure for the overall forecast goodness.

- Soil physical properties have a large spatial variability. Authors try to overcome this issue mainly with pedo-transfer functions. In my opinion, if the methodology shifts from an empirical approach to a physically based approach, the best option would be to switch to a probabilistic approach, in which e.g. a Montecarlo module is used to stochastically handle the inherent variability of soil properties. Have the Authors considered the pros and cons of a similar approach?

We agree that testing different parametrization ranges would be helpful at characterizing uncertainty of the model chain. We attempted this by testing different sources of soil properties information (SoilGrids, soil samples) and by testing extreme values of soil hydrological properties (4 different uniform-texture profiles). We also considered the use of probabilistic approaches to test different model parametrizations, an option which is implemented in the latest version of the CoupModel. However, the statistical analysis of the output of the soil moisture model (i.e. the time series of 133 sites with 11 modelling depths at hourly resolution) and the fitting of the statistical model (particularly the scanning of the different threshold values) is computationally very intensive. It was thus not possible to test the landslide forecast goodness of a soil moisture model with a probabilitstic approach. Nevertheless, we agree that this would be beneficial if it was done in a future study.